# Grouping Nodes with known Value Differences: A lossless UCT-based Abstraction Algorithm

**Robin Schmöcker**
Institute for Information Processing
Leibniz University Hannover
Hannover, Germany
schmoecker@tnt.uni-hannover.de

**Alexander Dockhorn**
SDU Metaverse Lab
University of Southern Denmark
Odense, Denmark
adoc@mmmi.sdu.dk

**Bodo Rosenhahn**
Institute for Information Processing
Leibniz University Hannover
Hannover, Germany
rosenhahn@tnt.uni-hannover.de

## Abstract

A core challenge of Monte Carlo Tree Search (MCTS) is its sample efficiency, which can be improved by grouping state-action pairs and using their aggregate statistics instead of single-node statistics. On the Go Abstractions in Upper Confidence bounds applied to Trees (OGA-UCT) is the state-of-the-art MCTS abstraction algorithm for deterministic environments that builds its abstraction using the Abstractions of State-Action Pairs (ASAP) framework, which aims to detect states and state-action pairs with the same value under optimal play by analysing the search graph. ASAP, however, requires two state-action pairs to have the same immediate reward, which is a rigid condition that limits the number of abstractions that can be found and thereby the sample efficiency. In this paper, we break with the paradigm of grouping value-equivalent states or state-action pairs and instead group states and state-action pairs with possibly different values as long as the difference between their values can be inferred. We call this abstraction framework Known Value Difference Abstractions (KVDA), which infers the value differences by analysis of the immediate rewards and modifies OGA-UCT to use this framework instead. The modification is called KVDA-UCT, which detects significantly more abstractions than OGA-UCT, introduces no additional parameter, and outperforms OGA-UCT on a variety of deterministic environments and parameter settings.

## 1 Introduction

Research into non-learning-based decision-making algorithms such as Monte Carlo Tree Search (MCTS) (Browne et al., 2012; Kocsis & Szepesvári, 2006) is an active field. On the one hand MCTS can be used for applications where a general on-the-fly applicable decision-making algorithm is needed such as Game Studios which rarely use Machine Learning (ML) based AI as they would have to be retrained whenever the game rules are modified (e.g. during development or patches). And on the other hand, though not the scope of this paper, foundational work in MCTS might potentially translate to improvements of ML algorithms such as Alpha Zero (Silver et al., 2017) that are built on MCTS.

One way to improve MCTS is to reduce the search space by grouping states and actions in the current MCTS search tree to enable an intra-layer information flow (Jiang et al., 2014; Anand et al., 2015; 2016) by averaging the visits and returns of all abstract action nodes in the same abstract node used for the Upper Confidence Bounds (UCB) formula in the tree policy, which increases the sample

efficiency. One key strength of one of the state-of-the-art abstraction algorithms On the Go Abstractions in Upper Confidence bounds applied to Trees (OGA-UCT) (Anand et al., 2016) is its exactness in the sense that if OGA-UCT groups two state-action pairs in a search tree where all possible successors of each state-action pair have been sampled, only state-action pairs are grouped that have the same $Q^*$ value, i.e., they have the same value under subsequent optimal play. This exactness condition, however, comes at the cost that state or state-action pairs that only differ slightly in their $Q^*$ value cannot be detected. This issue was slightly alleviated with the introduction of $(\varepsilon_a, \varepsilon_t)$-OGA (Schmöcker et al., 2025d), which is equivalent to $(\varepsilon_a, 0)$-OGA in deterministic environments that allows for small errors in the immediate reward or transition function when building an abstraction. However, this also brings two downsides with it. Firstly, it introduces two parameters, which makes tuning harder, and secondly, grouping non-value equivalent state-action pairs might even be harmful to the performance, as they can make convergence to the optimal action impossible.

In this work we propose Known Value Differences Abstractions UCT (KVDA-UCT), that relaxes the strict abstraction conditions of OGA-UCT to detect almost as many abstractions as $(\varepsilon_a, 0)$-OGA in deterministic environments but without losing the exactness condition. The novel idea that makes this possible is to deliberately group states or state-action pairs that do not have the same value if we know the difference between their values which is inferred by analysis of the search tree's immediate rewards. When the abstractions are used, instead of averaging state-action pair values directly, their difference-accounted values are averaged. The contributions of this paper can be summarized as follows:

**1.** We introduce the Known-Value-Difference (KVDA) abstraction framework that extends the Abstractions of State-Action Pairs (ASAP) framework used by OGA-UCT. Fig. 1 is an example of a simple state-transition graph in which KVDA finds three non-trivial abstractions, while ASAP would detect none.

**2.** We propose and empirically evaluate KVDA-UCT, a modification of OGA-UCT that introduces no parameters and uses and builds KVDA abstractions. The introduction of KVDA-UCT aims to address the abstraction literature gap for deterministic settings since existing modifications of OGA-UCT have only focused on stochastic settings (Anand et al., 2016; Xu et al., 2023; Schmöcker et al., 2025d;c). We show that KVDA-UCT outperforms OGA-UCT in most of the here-considered deterministic environments. We also compare KVDA-UCT with $(\varepsilon_a, 0)$-OGA and show that it either performs equally well or better than a parameter-optimized $(\varepsilon_a, 0)$-OGA agent.

**3.** Though not the main focus of this paper, we also consider the stochastic setting where we generalize KVDA-UCT to $\varepsilon_t$-KVDA which allows for errors in the transition function when building the abstraction. Furthermore, we compare it to $(\varepsilon_t, \varepsilon_a)$-OGA and show that, unlike in the deterministic setting, KVDA rarely performs better than $(\varepsilon_t, \varepsilon_a)$-OGA in stochastic environments. Still, there are some environments in which $\varepsilon_t$-KVDA achieves the best performances, which allows one to view it as an additional tool to improve the parameter-optimized performance.

The paper is structured as follows. In **Section** 2, the theoretical groundwork for automatic state and state-action pair abstractions is laid. In particular, we define OGA-UCT and $(\varepsilon_a, \varepsilon_t)$-OGA. Then, in **Section** 3, our novel KVDA framework and both KVDA-UCT and $\varepsilon_t$-KVDA are introduced. Afterwards, in **Section** 4, the experimental setup is defined, then in **Section** 5, the experimental results using this setup are shown and discussed. The paper is concluded by a discussion of the limitations of KVDA-UCT and avenues for future work in **Section** 6.

## 2 FOUNDATIONS OF AUTOMATIC ABSTRACTIONS

In this section, we will be laying the theoretical groundwork for this paper as well as introducing related work. For a comprehensive overview of non-learning-based abstractions, we refer to the survey paper by Schmöcker and Dockhorn (Schmöcker & Dockhorn, 2025).

**Problem model and optimization objective:** For our purposes, finite Markov Decision Processes (Sutton & Barto, 2018) are used as the model for sequential, perfect-information decision-making tasks. $\Delta(X)$ denotes the probability simplex of a finite, non-empty set $X$ and the power set of $X$ is denoted by $\mathcal{P}(X)$.

*Definition*: An *MDP* is a 6-tuple $(S, \mu_0, T, \mathbb{A}, \mathbb{P}, R)$ where the components are as follows:

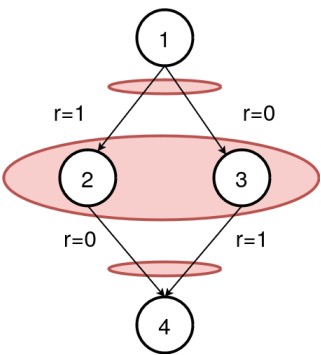

Figure 1: An example of an MDP state-transition graph where the state-of-the-art abstraction framework ASAP (Anand et al., 2015) would detect no abstractions while **our method Known-Value-Difference-Abstractions (KVDA)** detects three non-trivial abstractions. In this example, circles represent states, arrows represent deterministic state-transitions and arrow annotations denote the immediate transition reward. All actions or states that are intersected by a red ellipse will be abstracted by KVDA.

- $S \neq \emptyset$ is the finite set of states.
- $\mu_0 \in \Delta(S)$ is the probability distribution for the initial state.
- $\emptyset \neq T \subsetneq S$ is the (possibly empty) set of terminal states.
- $\mathbb{A} \colon (S \setminus T) \mapsto A$ maps each state $s$ to the available actions $\emptyset \neq \mathbb{A}(s) \subseteq A$ at state $s$ where $|A| < \infty$.
- $\mathbb{P} \colon (S \setminus T) \times A \mapsto \Delta(S)$ is the stochastic transition function where $\mathbb{P}(s' \mid s, a)$ is used to denote the probability of transitioning from $s \in (S \setminus T)$ to $s' \in S$ after taking action $a \in \mathbb{A}(s)$ in $s$.
- $R \colon (S \setminus T) \times A \mapsto \mathbb{R}$ is the reward function.

Let $M = (S, \mu_0, T, \mathbb{A}, \mathbb{P}, R)$ be an MDP. We define $P \coloneqq \{(s, a) \mid s \in (S \setminus T), a \in \mathbb{A}(s)\}$ as the set of all legal state-action pairs. The objective is to find a mapping (i.e. an agent) $\pi \colon S \mapsto \Delta(A)$ such that $\pi$ maximizes the expected episode's return where the (discounted) return of an episode $s_0, a_0, r_0, \ldots, s_n, a_n, r_n, s_{n+1}$ with $s_{n+1} \in T$ is given by $\gamma^0 r_0 + \ldots + \gamma^n r_n$.

**Abstractions of State-Action Pairs (ASAP):** For MCTS-based abstraction research, the goal has been to detect state-action pairs with the same $Q^*$ value (the value under subsequent optimal play) in the search graph to increase sample efficiency by an intra-layer information flow (Jiang et al., 2014; Anand et al., 2015; 2016). In general, by abstractions of either the states of state-action-pairs we refer to equivalence relations over the state set $S$ or state-action pair set $P$. The equivalence classes are abstract states or state-action pairs.

The current state of the art is the Abstraction of State-Action Pairs in UCT (ASAP) abstraction framework (Anand et al., 2015) that proposes rules to detect value-equivalent states and state-action pairs given an MDP transition graph and applies it to the current MCTS search graph (for details on MCTS, see Section A.8). The core idea of ASAP is to alternatingly construct a state abstraction given a state-action pair abstraction and a state-action pair abstraction given a state abstraction.

Assume one is given a state abstraction $\mathcal{E}' \subseteq S \times S$. The corresponding ASAP state-action pair abstraction $\mathcal{H} \subseteq P \times P$ is defined as grouping those state-action pairs with the same immediate reward and equal abstract successor distribution. Concretely, any state-action-pair $(s_1, a_1), (s_2, a_2)$ is equivalent i.e. $((s_1, a_1), (s_2, a_2)) \in \mathcal{H}$ if and only if

$$F_a \coloneqq |R(s_1, a_1) - R(s_2, a_2)| = 0$$

$$F_t \coloneqq \sum_{x \in \mathcal{X}} \left| \sum_{s' \in x} \mathbb{P}(s' \mid s_1, a_1) - \mathbb{P}(s' \mid s_2, a_2) \right| = 0, \tag{1}$$

where $\mathcal{X}$ are the equivalence classes of $\mathcal{E}'$. And given a state-action pair abstraction $\mathcal{H}' \subseteq P \times P$, the corresponding ASAP state abstraction $\mathcal{E}$ groups all states whose actions can be mapped to each

other, concretely:

$$(s_1, s_2) \in \mathcal{E} \iff$$
$$\forall a_1 \in \mathbb{A}(s_1) \, \exists a_2 \in \mathbb{A}(s_2) : ((s_1, a_1), (s_2, a_2)) \in \mathcal{H}_{i+1} \quad (2)$$
$$\forall a_2 \in \mathbb{A}(s_2) \, \exists a_1 \in \mathbb{A}(s_1) : ((s_1, a_1), (s_2, a_2)) \in \mathcal{H}_{i+1}.$$

To obtain the ASAP abstraction for a given MDP, these two constructing steps are repeated alternatingly until convergence. Note that the ASAP abstraction differs from MDP bisimulations as well as MDP homomorphisms (Ravindran & Barto, 2004) because state-action pairs can abstracted even if their parent states are not abstracted.

**OGA-UCT:** Anand et al. (2016) proposed OGA-UCT, which builds an ASAP-like abstraction in parallel to running MCTS. When building the abstraction OGA starts with the initial state abstraction that groups all terminal states of the same layer and puts the remaining states in their own singleton equivalence class. Furthermore, when building the ASAP abstraction on the current search graph, OGA ignores non-yet-sampled successors of state-action pairs that appear in Equation 2. To make the frequent recomputation of the ASAP abstraction feasible, OGA keeps track of a recency counter for each Q-node and once it surpasses a certain threshold, recomputes its abstraction. If the abstraction changed, the parent states are recomputed too (and possibly their Q-node parents if their abstraction changed). By only locally checking for errors in the abstraction, OGA is able to keep track of an ASAP-like abstraction that is always close to the true ASAP abstraction of the current search tree.

The only MCTS component that OGA-UCT affects is the tree policy which is enhanced by using the aggregate returns and visits of a Q-node's abstract node to enhance the UCB value.

$(\varepsilon_a, \varepsilon_t)$**-OGA**: The ASAP framework groups only value equivalent states and state-action pairs. This condition can be relaxed like already done by a predecessor of OGA-UCT, called AS-UCT (Jiang et al., 2014), by allowing the rewards in Equation 1 to differ by some threshold $\varepsilon_a > 0$ and the transition error $F_t$ of Equation 1 to lie in the interval $[0, \varepsilon_t]$ where $0 \leq \varepsilon_t \leq 2$ is another parameter *which does not have any effect in deterministic environments*. Since in general, positive threshold values do not induce an equivalence relation over state-action pairs, OGA-UCT has to be slightly modified to accommodate these approximate abstractions. This has been done by Schmöcker and Dockhorn (Schmöcker et al., 2025d) who introduced $(\varepsilon_a, \varepsilon_t)$-OGA.

**Auther automatic abstraction algorithm:** The ASAP framework is the direct successor of Abstraction of States (AS) by Jiang et al. (2014) that abstracts states if and only if their actions are pairwise similar in the sense that Equation 2 only has to be approximately satisfied as described in the $(\varepsilon_a, \varepsilon_t)$-OGA section above. While this paper focuses on deterministic domains, $(\varepsilon_a, \varepsilon_t)$-OGA (and related algorithms) have been further improved for stochastic settings by defining intra-abstraction policies (Schmöcker et al., 2025c), or dynamically abandoning the abstraction (Xu et al., 2023; Schmöcker et al., 2025d).

Yet another paradigm is demonstrated by Hostetler et al. (2015) and Schmöcker et al. (2025a) who optimistically construct abstractions by starting with a very coarse abstraction (e.g. grouping everything together) and then they refine this abstraction, for example by repeatedly splitting large node groups in half (Hostetler et al., 2015).

Research effort has also been dedicated towards automatic abstractions of the transition function, which on an abstract level can be described as pruning certain successors from the transition function (Sokota et al., 2021; Yoon et al., 2008; 2007; Saisubramanian et al., 2017).

## 3 METHOD

This section introduces our novel KVDA-UCT algorithm which is designed to detect strictly more abstractions than OGA-UCT which allows one to compute more accurate UCB values during both the decision and tree policy which in turn leads to performance improvements. First, we will provide an example of a concrete search graph in which ASAP misses abstractions that the Known Value Differences Abstractions (KVDA) framework would find which will be introduced after the search graph example. At the end of this section, we will be describing how KVDA is integrated into UCT to yield KVDA-UCT.

### 3.1 WHICH ABSTRACTIONS ASAP MISSES

Consider the state-transition graph that is illustrated in Fig. 1 that consists of four states and four actions. The ASAP framework would not detect any equivalences. In fact, any framework that aims at finding value-equivalent states or state-action pairs would at most be able to detect that the two root actions are value-equivalent. However, by analysing the state graph, one could derive that the $Q^*$ values of the action of node 2 differs only by 1 from the $Q^*$ value of the action of node 3. This holds true even if the state-graph would extend past node 4. Consequently, the $V^*$ values of nodes 2 and 3 differ only by 1. This in turn implies that the two actions of node 1 must have the same $Q^*$ value.

The analysis of this example can be further generalized which is the core idea of **Known-Value-Difference-Abstractions (KVDA)** framework which is to abstract/group states and state-action pairs if one can derive their value differences. Intuitively, KVDA abstractions are identical to ASAP abstractions except that immediate reward errors are ignored and accounted for by tracking them as value differences. Later, when using these abstractions to enhance MCTS, the differences only have to be subtracted when aggregating values. When viewed from the lenses of the KVDA framework, ASAP only groups states or state-action pairs with a value difference of 0, i.e. detects only true equivalences.

While this was only a toy example of illustration purposes, the KVDA framework does also discover more abstractions in practice as will later be shown experimentally. A concrete example of an MDP in which KVDA detects strictly more abstractions than ASAP is Game of Life (see Section A.9). In this environment, the immediate reward is also equal to the number of living cells. Therefore, state-action pairs from states with different number of living cells cannot be ASAP-abstracted even though they might have the successor(s). In contrast, KVDA is able discover such abstractions.

Next, we will formalize the KVDA framework which both in theory and in our empirical evaluations (see Tab. 1, more details are given in the experimental section) detects strictly more abstractions.

### 3.2 THE KNOWN-VALUE-DIFFERENCE-ABSTRACTIONS FRAMEWORK

Our method, the Known-Value-Difference-Abstractions (KVDA) extends the ASAP definition by additionally grouping states or state-action-pairs whose $V^*$ or $Q^*$ difference is known. While ASAP iteratively builds abstractions on abstractions, KVDA bootstraps of an abstraction, difference-function pair. More concretely, given a state abstraction $\mathcal{E}'$ (or a state-action pair abstraction $\mathcal{H}'$) and a difference function $d_s' \colon S \times S \mapsto \mathbb{R}$ (or $d_a' \colon P \times P \mapsto \mathbb{R}$), both a state-action-pair abstraction $\mathcal{H}$ (or a state abstraction $\mathcal{E}$) is produced as well as a state-action pair difference function $d_a \colon P \times P \mapsto \mathbb{R}$ (or a state difference function $d_s \colon S \times S \mapsto \mathbb{R}$). Both $d_a$ and $d_s$ will be constructed such that

$$d_a(p_1, p_2) = d_a^*(p_1, p_2) := Q^*(p_2) - Q^*(p_1)$$
$$d_s(s_1, s_2) = d_s^*(s_1, s_2) := V^*(s_2) - V^*(s_1)$$

(3)

for any pair of states or state-action pairs in the same equivalence class (i.e., in the same abstract state). Even though in the experimental section, we will mostly consider deterministic environments, the now-to-be-described KVDA framework will be applicable to any MDP, including stochastic ones. Next, starting with the base case, we will formalize how KVDA abstractions are built.

**The base case:** Like ASAP, KVDA groups all terminal states into the same initial abstract node, the remaining states are singleton abstract nodes. The difference function between all nodes in the terminal abstract node is initialized with 0.

**State-action-pair abstractions:** Using the same notation as in Section 2, let $\mathcal{E}' \subseteq S \times S$ be a state abstraction and $d_s' \colon S \times S \mapsto \mathbb{R}$ be a difference function. The corresponding KVDA state-action pair abstraction $\mathcal{H} \subseteq P \times P$ is defined as follows: Any state-action pair $(s_1, a_1), (s_2, a_2)$ is equivalent i.e. $((s_1, a_1), (s_2, a_2)) \in \mathcal{H}$ if and only if

$$\sum_{x \in \mathcal{X}} \left| \sum_{s' \in x} \mathbb{P}(s' \mid s_1, a_1) - \mathbb{P}(s' \mid s_2, a_2) \right| = 0,$$

(4)

where $\mathcal{X}$ are the equivalence classes of $\mathcal{E}'$. Note that this is almost identical to the ASAP definition except that the immediate rewards do not have to coincide. The difference function $d_a \colon P \times P \mapsto \mathbb{R}$ for two $(p_1, p_2) \in \mathcal{H}$ is given by

$$
\begin{aligned}
d_a(p_1, p_2) = &R(p_2) - R(p_1) \\
&+ \sum_{x \in \mathcal{X}} \sum_{s' \in x} \left( \mathbb{P}(s' \mid p_1) - \mathbb{P}(s' \mid p_2) \right) \cdot d_s'(s', s_x)
\end{aligned}
\tag{5}
$$

where for each $x \in \mathcal{X}$, $s_x$ is an arbitrarily chosen but fixed representative of $\mathcal{X}$. We will later see that the value of $d_a$ is independent of this choice.

**State abstractions:** Given a state-action pair abstraction $\mathcal{H}' \subseteq P \times P$ and a state-action pair difference function $d_a \colon P \times P \mapsto \mathbb{R}$ the corresponding KVDA state abstraction $\mathcal{E} \subseteq S \times S$ groups all states whose actions can be mapped to each other, and whose mappings all have the same value difference:

$$
\begin{aligned}
(s_1, s_2) \in \mathcal{E} \iff \exists d \in \mathbb{R} : \\
\forall a_1 \in \mathbb{A}(s_1)\, \exists a_2 \in \mathbb{A}(s_2) : ((s_1, a_1), (s_2, a_2)) \in \mathcal{H}' \\
\wedge d_a((s_1, a_1), (s_2, a_2)) = d \\
\forall a_2 \in \mathbb{A}(s_2)\, \exists a_1 \in \mathbb{A}(s_1) : ((s_1, a_1), (s_2, a_2)) \in \mathcal{H}' \\
\wedge d_a((s_2, a_2), (s_1, a_1)) = -d.
\end{aligned}
\tag{6}
$$

The difference function $d_s(s_1, s_2)$ is defined as the value $d$ in the equation above.

**Theoretical guarantees:**

*Convergence:* Given an MDP, the above-described construction steps can be repeated until convergence, which is guaranteed as in our MDP definition, there are finitely many states and state-action pairs and each construction step either leaves the abstraction unchanged or reduces the number of equivalence classes. The abstraction that one obtains at convergence is called the KVDA abstraction of an MDP.

*Soundness of $d_a$ and $d_s$:* Both $d_a$ and $d_s$ at convergence are equal to the differences of their arguments $Q^*$ or $V^*$ values as formulated in Eq. 3. This is proven in the supplementary materials in Section A.2.

## 3.3 KVDA-UCT AND $\varepsilon_\text{T}$-KVDA

In this section, we will describe how the KVDA abstraction framework is integrated into MCTS, which is fully analogous to how $(\varepsilon_a, \varepsilon_t)$-OGA (which itself is equivalent to OGA-UCT for $\varepsilon_a = \varepsilon_t = 0$) integrates the ASAP framework. The usage of the following to-be-described modifications to $(\varepsilon_a, \varepsilon_t)$-OGA is called $\varepsilon_\text{t}$-**KVDA**. For the case $\varepsilon_t = 0$, the algorithm is simply called **KVDA-UCT**. $\varepsilon_\text{t}$-**KVDA** does not depend on $\varepsilon_a$ unlike $(\varepsilon_a, \varepsilon_t)$-OGA. The detailed pseudocode for this method is provided in Pseudocode 1 in which the differences to $(\varepsilon_a, \varepsilon_t)$-OGA are shaded in blue.

**1)** Instead of (approximate) ASAP abstractions, (approximate) KVDA abstractions of the search tree are built. Both $(\varepsilon_a, \varepsilon_t)$-OGA and $\varepsilon_\text{t}$-KVDA allow the transition error $F_t$ of Equation 1 to be in the interval $[0, \varepsilon_t]$.

**2)** For efficient distance function calculations, each abstract node keeps track of a representative. Value differences are only calculated with respect to the representative. An abstract node's representative is the first original node added to the abstract node, and if that one is removed, a random new representative is chosen.

**3)** Each abstract Q node's value is tracked as the value of its representative $\mathcal{Q}_\text{R}$ that encodes the state-action-pair $p_\text{R} \in P$. When another original node $\mathcal{Q}$ of the same abstract node representing $p \in P$ is backed up in the MCTS backup phase with value $v \in \mathbb{R}$, the value $v + d_a(p, p_\text{R})$ is added to the abstract node's statistics. In turn, $\mathcal{Q}$ extracts its aggregated returns from the abstract node by subtracting $d_a(p, p_\text{R})$ from the abstract node's value. If an abstract node's representative changes to $\mathcal{Q}_\text{R}'$ encoding $p_\text{R}' \in P$, then $n \cdot d_a(p_\text{R}, p_\text{R}')$ is added to the statistics where $n$ is the abstract visit count.

Note that in the deterministic setting where $d_a$ is always computed correctly, this implies that KVDA-UCT will never visit state-action pairs as long as they are abstracted with another state-action pair with a positive $d_a$ difference (i.e. when they are provably not the optimal action) since these state-action pairs will have the same exploration term and Q values that differ by $d_a$. This is a strength that OGA-UCT does not possess.

**4)** To reduce the computational load of finding a perfect match as the one required for the state abstractions in Equation 6, we check for a stricter condition for states $s_1, s_2$. It is first checked that within $s_1$ (and analogously $s_2$) all actions within the same abstract node have a value difference of zero. Then, it is tested for all abstract Q nodes of $s_1$ if the value difference between an arbitrarily chosen ground action of $s_1$ and one of $s_2$ is constant for all abstract nodes.

**5)** Since the KVDA abstraction also depends on the difference functions, whenever a Q-node's recency counter reaches the threshold, the value difference to its representative is recalculated. A change in this difference also results in a reevaluation (and subsequent recency counter reset) of the parent nodes' abstractions and difference functions.

## 4 EXPERIMENT SETUP

In this section, we describe the general experiment setup. Any deviations from this setup will be explicitly mentioned.

**Problem models:** For this paper, we ran our experiments on a variety of MDPs, all of which are either from the International Probabilistic Planning Conference (Grzes et al., 2014), are well-known board games, or are commonly used in the abstraction algorithm literature. Since we will compare KVDA-UCT to OGA-UCT we chose only environments with a non-constant and non-sparse reward function, as constant reward environments would imply that KVDA-UCT and OGA-UCT are semantically equivalent.

All MDPs originally feature stochasticity, however, for some experiments we considered their deterministic versions which are obtained by sampling a single successor of each state-action pair. For each episode, new successors are sampled. All experiments were run on the finite horizon versions of the considered MDPs with a default horizon of 50 steps and 200 for the board games with a planning horizon of 50 and a discount factor $\gamma = 1$.

The board games are transformed into MDPs by inserting a deterministic One-Step-Lookahead agent as the opponent. Furthermore, since they are all sparse-reward (i.e. either win or lose) they were also transformed into dense-reward MDPs by using heuristics. For each board game we defined a heuristic $V^h$ that assigns each state a heuristic value for its state value. The reward of the (deterministic) transition $s, a, s'$ is then given by $V^h(s') - V^h(s)$. The concrete heuristics used along with a brief description of all MDPs is provided in the supplementary materials in Section A.9. The deterministic MDPs are denoted by adding the prefix $d$- and the stochastic ones are denoted by the prefix $s$-.

**Parameters:** Since the problem domains have vastly different reward scales, we use the dynamic exploration factor Global Std (Schmöcker et al., 2025e) which has the form $C \cdot \sigma$ where $\sigma$ is the standard deviation of the Q values of all nodes in the search tree and $C \in \mathbb{R}^+$ is some fixed parameter. Furthermore, we always use $K = 3$ as the recency counter which was proposed by Anand et al. (2016).

**Evaluation:** Each experiment is repeated at least 2000 times and all confidence that we denote in the following are $99\%$ confidence intervals with range $\approx 2 \cdot 2.33$ times the standard error.

**Normalized pairings score**: We will later construct the *normalized pairings score* to test the generalization capabilities of KVDA-UCT. This is the same score as used in Schmöcker et al. (2025e). The pairings score is constructed as follows. Let $\{\pi_1, \ldots, \pi_n\}$ be $n$ agents (e.g. each KVDA-UCT agent along with its parameter setting is an agent) where each agent was evaluated on $m$ tasks (later, a task will be a given MCTS iteration budget and an environment). This induces a matrix of size $n \times n$ with the entry $(i, j)$ being equal to the number of tasks where $\pi_i$ achieved a higher performance than $\pi_j$ subtracted by the number of times it performed worse, divided by $m$. The normalized pairings score $s_i \leq i \leq n$ is then obtained by taking the average of $i$-th row when excluding the $i$-th column.

**Reproducibility:** For reproducibility, we released our implementation (Schmöcker, 2025). Our code was compiled with g++ version 13.1.0 using the -O3 flag (i.e. aggressive optimization).

## 5 EXPERIMENTS

This section presents the experimental results of KVDA-UCT. We considered two settings. Firstly, we measured the number of additional abstractions KVDA-UCT finds in comparison to OGA-UCT. Then, we evaluated KVDA-UCT on deterministic settings in which the losslessness of the abstraction is guaranteed. Lastly, we present the results for stochastic environments.

**Abstractions that KVDA-UCT finds but OGA-UCT does not:** Firstly, we empirically measured the number of non-trivial abstractions (i.e. those that are not of size one) that KVDA-UCT, OGA-UCT, and $(\infty, 0)$-UCT find. For all deterministic environments, Tab. 1 denotes the average ratio of non-trivial abstract state-action pairs (synonymously abstract Q nodes) to the number of total abstract Q nodes in their respective search trees. In most environments, KVDA-UCT detects more abstractions than OGA-UCT, including environments where the abstraction rate more than doubles, such as SysAdmin or Wildfire. Furthermore, KVDA-UCT detects roughly as many abstractions as $(\infty, 0)$-OGA in most environments, which fully ignores rewards when building abstractions. Of course, finding more abstractions is not a difficult task in of itself, since could just group every node by default. However, both ASAP and the KVDA abstractions are lossless in the deterministic setting, i.e. they group only state-action pairs with identical or known-difference $Q^*$ values. We empirically verified this for those environments which could be solved with value iteration, which are Saving, Sailing Wind, and Skills Teaching. Hence, KVDA is able to detect strictly more true equivalences than ASAP.

Table 1: The average ratio of abstract Q nodes in KVDA, OGA-UCT's and $(\infty, 0)$-OGA's respective search trees after 1000 iterations with $\lambda = 2$ that encompass more than one original node divided by the total number of abstract Q nodes. The states in which these search trees are built are sampled from an OGA-UCT agent with $\lambda = 2$ and 500 iterations. This measurement excludes all size-one abstract Q nodes whose original Q node has not yet reached the recency counter (see Section 2) required for the first abstraction update. Hence, a ratio of 1 means that no abstractions were found while a ratio of 0 means that every state-action pair has been abstracted with another. Note that KVDA-UCT (our method) finds more abstractions than OGA-UCT in nearly every environment and in general as many as $(\infty, 0)$-OGA. While there are some environments where there is no gain, such as Constrictor, there are other environments such as Wildfire or SysAdmin, where the abstraction rate more than doubles.

| Domain | KVDA-UCT (ours) | OGA-UCT | $(\infty, 0)$-OGA |
|---|---|---|---|
| d-Academic Advising | 0.69 | 0.73 | 0.68 |
| d-Connect4 | 0.80 | 0.91 | 0.80 |
| d-Constrictor | 0.98 | 0.97 | 0.97 |
| d-Cooperative Recon | 0.44 | 0.56 | 0.48 |
| d-Earth Observation | 0.65 | 0.99 | 0.68 |
| d-Elevators | 0.28 | 0.32 | 0.29 |
| d-Game of Life | 0.54 | 0.56 | 0.55 |
| d-Manufacturer | 0.64 | 0.95 | 0.79 |
| d-Othello | 0.96 | 0.99 | 0.96 |
| d-Pusher | 0.96 | 0.99 | 0.96 |
| d-Push Your Luck | 0.23 | 0.48 | 0.16 |
| d-Red Finned Blue Eye | 0.72 | 0.84 | 0.70 |
| d-Sailing Wind | 0.70 | 0.92 | 0.74 |
| d-Saving | 0.96 | 0.96 | 0.96 |
| d-Skills Teaching | 0.59 | 0.65 | 0.59 |
| d-SysAdmin | 0.15 | 0.48 | 0.20 |
| d-Tamarisk | 0.35 | 0.56 | 0.39 |
| d-Traffic | 0.53 | 0.54 | 0.53 |
| d-Wildfire | 0.19 | 0.37 | 0.30 |
| d-Wildlife Preserve | 0.06 | 0.08 | 0.08 |

**Parameter-optimized KVDA-UCT:** Next, we compared KVDA-UCT, $(0, 0)$-OGA (i.e. standard OGA-UCT), and $(\varepsilon_a, 0)$-OGA with $\varepsilon_a > 0$ in terms of their parameter-optimized performances on deterministic environments. For all methods, we optimized over $C \in \{0.5, 1, 2, 4, 8, 16\}$ and some domain-specific values for $\varepsilon_a$ that are listed in the supplementary materials in Tab. 4. Each parameter

Table 2: Average returns ($\uparrow$) using 100 (right) and 1000 (left) MCTS iterations for KVDA-UCT, $(\infty, 0)$-OGA, $(0, 0)$-OGA (i.e. OGA-UCT), and the maximal performance of $(\varepsilon_a, 0)$-OGA when varying $\varepsilon_a > 0$. In the supplementary materials in Tab.4 the domain-specific $\varepsilon_a$-values are listed. All performances are the maximal performances obtained by varying the exploration constants $C \in \{0.5, 1, 2, 4, 8, 16\}$. Note that KVDA-UCT (our method) outperforms OGA-UCT in most environments. Furthermore, KVDA-UCT is either equal or performs better than parameter-optimized $(\varepsilon_a, 0)$-OGA even though KVDA introduces no extra parameter.

| | 1000 Iterations | | | | 100 Iterations | | | |
|---|---|---|---|---|---|---|---|---|
| **Domain** | $(0,0)$-OGA | $(\infty,0)$-OGA | $(\varepsilon_a,0)$-OGA | **KVDA** (ours) | $(0,0)$-OGA | $(\infty,0)$-OGA | $(\varepsilon_a,0)$-OGA | **KVDA** (ours) |
| d-Connect4 | $42.7 \pm 0.6$ | $46.8 \pm 1.0$ | $47.5 \pm 0.9$ | $\mathbf{47.9 \pm 0.6}$ | $28.3 \pm 0.3$ | $\mathbf{28.8 \pm 0.5}$ | $\mathbf{28.8 \pm 0.5}$ | $28.4 \pm 0.4$ |
| d-Constrictor | $96.1 \pm 0.3$ | $95.0 \pm 0.2$ | $\mathbf{96.1 \pm 0.3}$ | $96.0 \pm 0.3$ | $85.0 \pm 0.6$ | $84.1 \pm 0.4$ | $84.9 \pm 0.9$ | $\mathbf{85.1 \pm 0.6}$ |
| d-Othello | $\mathbf{181.2 \pm 1.2}$ | $160.2 \pm 0.8$ | $180.7 \pm 1.5$ | $179.6 \pm 1.2$ | $154.7 \pm 1.3$ | $146.8 \pm 0.8$ | $\mathbf{155.1 \pm 1.8}$ | $154.8 \pm 1.3$ |
| d-Pusher | $\mathbf{104.5 \pm 0.0}$ | $\mathbf{104.5 \pm 0.0}$ | $\mathbf{104.5 \pm 0.0}$ | $\mathbf{104.5 \pm 0.0}$ | $\mathbf{104.5 \pm 0.0}$ | $\mathbf{104.5 \pm 0.0}$ | $\mathbf{104.5 \pm 0.0}$ | $\mathbf{104.5 \pm 0.0}$ |
| d-Academic Advising | $\mathbf{-39.9 \pm 0.2}$ | $-40.0 \pm 0.2$ | $-40.0 \pm 0.2$ | $-40.1 \pm 0.2$ | $-56.6 \pm 0.7$ | $-56.2 \pm 0.6$ | $-56.2 \pm 0.6$ | $\mathbf{-56.1 \pm 0.6}$ |
| d-Cooperative Recon | $16.1 \pm 0.1$ | $14.8 \pm 0.1$ | $\mathbf{16.2 \pm 0.1}$ | $16.0 \pm 0.1$ | $10.8 \pm 0.3$ | $10.9 \pm 0.3$ | $11.0 \pm 0.3$ | $\mathbf{11.0 \pm 0.3}$ |
| d-Earth Observation | $-7.18 \pm 0.11$ | $-30.0 \pm 0.4$ | $-30.0 \pm 0.4$ | $\mathbf{-7.02 \pm 0.10}$ | $-10.2 \pm 0.2$ | $-28.4 \pm 0.2$ | $-28.4 \pm 0.2$ | $\mathbf{-7.45 \pm 0.12}$ |
| d-Elevators | $-14.9 \pm 0.4$ | $-14.2 \pm 0.3$ | $-14.0 \pm 0.3$ | $\mathbf{-13.9 \pm 0.4}$ | $-18.0 \pm 0.3$ | $-18.1 \pm 0.3$ | $\mathbf{-17.9 \pm 0.3}$ | $-18.0 \pm 0.3$ |
| d-Game of Life | $\mathbf{666.7 \pm 2.5}$ | $664.6 \pm 2.5$ | $665.8 \pm 2.5$ | $664.8 \pm 2.6$ | $641.8 \pm 2.1$ | $\mathbf{642.1 \pm 2.2}$ | $\mathbf{642.1 \pm 2.2}$ | $641.9 \pm 2.1$ |
| d-Manufacturer | $-1255.6 \pm 15.0$ | $-1658.4 \pm 22.1$ | $-1246.0 \pm 16.2$ | $\mathbf{-1158.2 \pm 19.4}$ | $-1423.1 \pm 15.9$ | $-1680.3 \pm 22.4$ | $-1392.6 \pm 20.2$ | $\mathbf{-1233.5 \pm 23.2}$ |
| d-Push Your Luck | $125.1 \pm 1.9$ | $66.7 \pm 1.1$ | $132.4 \pm 2.5$ | $\mathbf{137.9 \pm 2.2}$ | $103.5 \pm 1.5$ | $66.3 \pm 0.9$ | $107.0 \pm 1.6$ | $\mathbf{107.6 \pm 1.5}$ |
| d-RedFinnedBlueEye | $8191.3 \pm 44.7$ | $7930.0 \pm 44.7$ | $7950.6 \pm 33.5$ | $\mathbf{8229.8 \pm 46.5}$ | $7683.4 \pm 33.8$ | $7464.4 \pm 34.7$ | $7495.1 \pm 35.1$ | $\mathbf{7698.2 \pm 34.0}$ |
| d-Sailing Wind | $-40.0 \pm 0.6$ | $-39.1 \pm 0.6$ | $-38.9 \pm 0.6$ | $\mathbf{-37.7 \pm 0.6}$ | $-64.7 \pm 0.8$ | $-64.9 \pm 0.9$ | $-64.9 \pm 0.9$ | $\mathbf{-63.6 \pm 0.8}$ |
| d-Saving | $\mathbf{66.0 \pm 0.2}$ | $63.0 \pm 0.3$ | $65.4 \pm 0.2$ | $65.4 \pm 0.2$ | $57.1 \pm 0.2$ | $55.2 \pm 0.3$ | $56.8 \pm 0.2$ | $\mathbf{57.2 \pm 0.2}$ |
| d-Skills Teaching | $207.9 \pm 4.6$ | $211.3 \pm 5.1$ | $211.3 \pm 5.1$ | $\mathbf{216.2 \pm 4.5}$ | $159.0 \pm 3.9$ | $158.3 \pm 3.9$ | $160.7 \pm 3.8$ | $\mathbf{162.3 \pm 3.8}$ |
| d-SysAdmin | $477.1 \pm 1.5$ | $448.4 \pm 1.3$ | $450.7 \pm 1.1$ | $\mathbf{477.2 \pm 1.5}$ | $475.5 \pm 1.7$ | $449.5 \pm 1.2$ | $450.1 \pm 1.2$ | $\mathbf{479.1 \pm 1.4}$ |
| d-Tamarisk | $-214.5 \pm 3.9$ | $-208.2 \pm 2.3$ | $-206.4 \pm 2.7$ | $\mathbf{-185.6 \pm 2.0}$ | $-315.8 \pm 6.5$ | $-285.4 \pm 4.8$ | $-284.9 \pm 4.7$ | $\mathbf{-263.0 \pm 4.8}$ |
| d-Traffic | $\mathbf{-1.54 \pm 0.07}$ | $-1.61 \pm 0.07$ | $-1.61 \pm 0.07$ | $-1.58 \pm 0.07$ | $\mathbf{-9.21 \pm 0.22}$ | $\mathbf{-9.21 \pm 0.22}$ | $\mathbf{-9.21 \pm 0.22}$ | $\mathbf{-9.21 \pm 0.22}$ |
| d-Wildfire | $-195.6 \pm 55.9$ | $-503.5 \pm 36.1$ | $-415.0 \pm 36.1$ | $\mathbf{-179.9 \pm 49.2}$ | $-194.1 \pm 36.1$ | $-498.7 \pm 36.1$ | $-408.9 \pm 36.1$ | $\mathbf{-173.2 \pm 36.1}$ |
| d-Wildlife Preserve | $1388.0 \pm 0.8$ | $1387.9 \pm 0.7$ | $1387.9 \pm 0.7$ | $\mathbf{1388.5 \pm 0.7}$ | $\mathbf{1388.9 \pm 0.9}$ | $\mathbf{1388.9 \pm 0.9}$ | $\mathbf{1388.9 \pm 0.9}$ | $\mathbf{1388.9 \pm 0.9}$ |

combination was evaluated on all here-considered deterministic environments with a budget of 100, 200, 500, and 1000 MCTS iterations. Tab. 2 list the performance for 100 and 1000 iterations, the tables for 200 and 500 iterations are found in the supplementary materials in Tab. 7 and 6. The results clearly show that KVDA either outperforms all other competitor methods or is tightly within the confidence bounds. There are a number of environments such as Manufacturer, Tamarisk, or Push Your Luck where KVDA-UCT simultaneously outperforms all competitor methods at once. Figure 2 plots the performances in dependence of the iteration budget for these tasks. The performance graphs for all other environments are found in the supplementary materials in Section A.7.

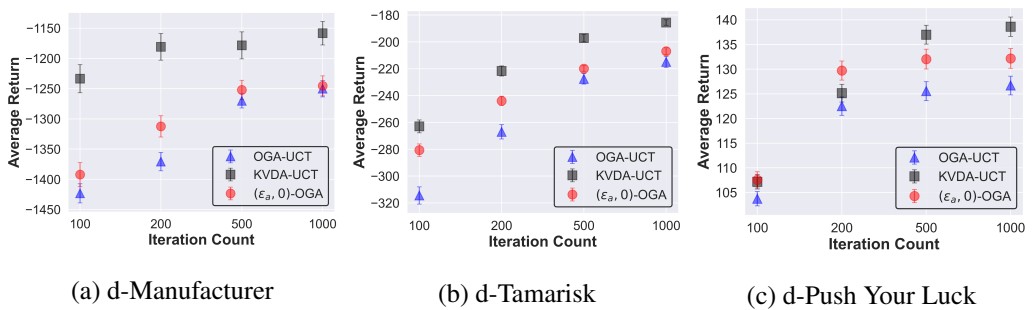

(a) d-Manufacturer       (b) d-Tamarisk       (c) d-Push Your Luck

Figure 2: The performance plots in dependence of the iteration budget of parameter-optimized KVDA-UCT (our method), OGA-UCT (Anand et al., 2016), and $(\varepsilon_a, 0)$-OGA on selected environments. KVDA-UCT clearly outperforms its competitors. The plots for the remaining environments can be found in the supplementary materials in Section A.7.

**Single-parameter KVDA-UCT:** Next, the generalization capabilities of KVDA-UCT in comparison to OGA-UCT and $(\varepsilon_a, 0)$-OGA on deterministic environments were tested. Like last section, we ran all algorithms using 100, 200, 500, and 1000 MCTS iterations whilst varying $C \in \{0.5, 1, 2, 4, 8, 16\}$ and $\varepsilon_a$ with domain-specific values (see supplementary materials Tab. 4). Instead of considering the best-performances, we calculated the normalized pairings score for each parameter combination (except for $(\varepsilon_a, 0)$-OGA where we considered the maximum performance across all $\varepsilon_a > 0$ values a single parameter combination), constructed over the tasks which are the pairs of all environments and iteration budgets. Bar chart 3 shows the 6 agents with the highest score as well as the agent with the lowest score. Both the top spots are occupied by our KVDA-method,

followed by OGA-UCT and $(\varepsilon_a, 0)$-OGA. The best performing algorithm overall is KVDA-UCT with $C = 4$.

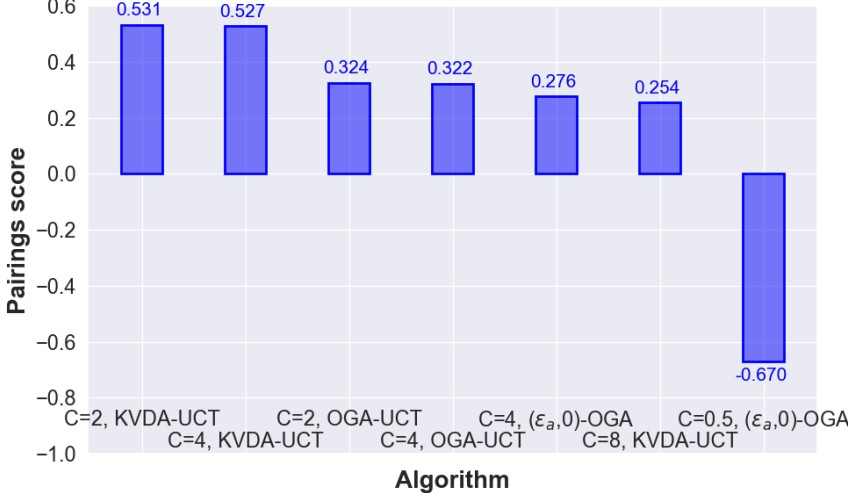

Figure 3: The normalized pairings score ($\uparrow$) for the top 6 and the worst agent on deterministic environments. The agents considered were KVDA-UCT (our method) which performs best overall, OGA-UCT (Anand et al., 2016), and $(\varepsilon_a, 0)$-OGA (Schmöcker et al., 2025d), $\varepsilon_a > 0$ with the exploration constants $C \in \{0.5, 1, 2, 4, 8, 16\}$ and budgets of $\{100, 200, 500, 1000\}$ iterations. The top two spots are occupied by our method KVDA-UCT, with the best overall performing algorithm being KVDA-UCT with $C = 2$.

$\varepsilon_t$**-KVDA on stochastic environments:** Lastly, the performance of $\varepsilon_t$-KVDA on stochastic environments were tested by running both $\varepsilon_t$-KVDA and $(\varepsilon_a, \varepsilon_t)$-OGA on the stochastic versions of the here-considered environments. We considered the iteration budgets of 100 and 1000 iterations and varied $C \in \{0.5, 1, 2, 4, 8, 16\}$, $\varepsilon_t \in \{0, 0.4, 0.8, 1.2, 1.6\}$ and the same domain-specific $\varepsilon_a$ that were used in the previous sections whose values are listed in the supplementary materials in Tab. 4. For each environment, Tab. 5 of the supplementary materials lists the parameter-optimized performances of both algorithms. In contrast to the deterministic setting, KVDA does not outperform OGA. For most environments both perform equally well with some exceptions such as Manufacturer were $\varepsilon_t$-KVDA clearly performs best and Tamarisk were $(\varepsilon_a, \varepsilon_t)$-OGA performs best. We believe mediocre performance of $\varepsilon_t$-KVDA stems from the fact that since $\varepsilon_t$-KVDA essentially ignores immediate rewards when it builds abstractions, the number of faulty abstractions that occur in this approximate setting is increased. In the deterministic setting, no faulty abstractions are built.

## 6 CONCLUSION, LIMITATIONS AND FUTURE WORK

This paper introduced KVDA-UCT, an extension to OGA-UCT, that additionally groups states and state-action pairs that do not have the same $Q^*$ or $V^*$ value as long as the difference is known. We evaluated and compared KVDA-UCT to OGA-UCT and $(\varepsilon_a, 0)$-OGA on a deterministic setting where KVDA-UCT outperforms all its competitors both in terms of generalization as well as the parameter-optimized performance. This does not hold true for the stochastic setting for which we generalized KVDA-UCT to $\varepsilon_t$-KVDA which is only clearly better than $(\varepsilon_a, \varepsilon_t)$-OGA in few environments.

A first avenue for future work will be to further investigate the reasons for the mediocre performance of the experiments in this paper of the stochastic setting and develop an extension of $\varepsilon_t$-KVDA suited for this setting. Another limitation of KVDA abstractions in its current form is its limitation to MDPs (i.e. single-player games). Even though MCTS is in principle applicable to multi-player games, KVDA is not because in any state, there is in general, no unique $V^*$ value from which differences can be built, as there are potentially many equilibria with different payoff vectors. Future work will be to extend KVDA to this setting.

## 7 REPRODUCIBILITY STATEMENT

In our experiment setup, we have a subsection called *Reproducibility* in which we provide a download link to the full codebase used for this project as well as compilation details. The codebase contains an elaborate README detailing the steps to reproduce the experiments.

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

# A SUPPLEMENTARY MATERIALS

## A.1 KVDA PSEUDOCODE

---

**Algorithm 1:** $\varepsilon_t$-KVDA

---

**Parameters:** *oga_args*
**Input:** *state*

1 **Globals:** *max_id, abstractionsQ, abstractionsStates, RecencyCount,* $d_a(\cdot,\cdot), d_s(\cdot,\cdot)$

2 *tree* = `init_tree`(*state*)

3 **for** $i = 1$ **to** *oga_args.iterations* **do**

4     `// For UCB the values` *abstractionsQ[Q].value* $+ d_a(Q, abstractionsQ[Q].representative)$ `are used`

    *leaf*, *path_to_leaf*, *newQnodes* = `treePolicy`(*tree*)

5     **foreach** *newQnode* $\in$ *newQnodes* **do**

6         *abstractionsQ[newQnode]* = `new_singleton_Q_abstraction`()

7         *abstractionsQ[newQnode].representative* = *newQnode*

8         *newQnode.id* = *max_id*++

9     **if** *leaf* $\notin$ *tree* **then**

10         *abstractionsStates[leaf]* = `new_singleton_state_abstraction`()

11         *abstractionsStates[leaf].representative* = *leaf*

12     `backup`(*path_to_leaf*,`rollout`(*leaf*))

13     **foreach** $(Q, s) \in$ *path_to_leaf* **do**

14         *abstractionsQ[Q].value* $+= d_a(Q, abstractionsQ[Q].representative)$

15

16     **foreach** $(Q, s) \in$ *path_to_leaf* **do**

17         `update_Q_abstraction`(*Q*)

18 **return** $\arg\max_a Q(state, a)$

19 **function** `update_Q_abstraction`(*Q*)

20     **if** *RecencyCount[Q]*++ $<$ *oga_args.K* **then**

21         **return**

22     *RecencyCount[Q]* = 0

23     *new_abs* = `compute_Q_abstraction`(*Q*)

24     *old_dist* = $d_a(Q, abstractionsQ[Q].representative)$

25     update $d_a(Q, new\_abs.representative)$ using Equation 5

26     **if** *abstractionsQ[Q]* $\neq$ *new_abs* **or** $d_a(new\_abs.representative, Q) \neq$ *old_dist* **then**

27         *abstractionsQ[Q].visits* $-=$ *Q.visits*; *new_abs.visits* $+=$ *Q.visits*

28         *abstractionsQ[Q].value* $-=$ *Q.value* $+ Q.visits \cdot d_s(Q, abstractionsQ[Q].representative)$

29         *new_abs.value* $+=$ *Q.value* $+ Q.visits \cdot d_s(Q, new\_abs.representative)$

30         **if** $Q ==$ *abstractionsQ[Q].representative* **and** *abstractionsQ[Q].size* $> 1$ **then**

31             $Q_{new}$ = `choose_new_representative_randomly`(*abstractionsQ[Q]*, excluding=*Q*)

32             recalculate affected $d_a$ values

33             *abstractionsQ[Q].value* $+=$ *abstractionsQ[Q].visits* $\cdot d_a(Q, Q_{new})$

34         *abstractionsQ[Q]* = *new_abs*

35         `update_state_abstraction`(*s*) // $Q = (s, a)$

36

---

---

**Algorithm 1:** $\varepsilon_t$-KVDA (continued)

---

48 **function** update_state_abstraction(*state*)
49    *new_abs* = compute_state_abstraction(*state*)                  // abstracts two states iff
     get_distance($s_1, s_2$) $\neq \infty$
50    *old_dist* = $d_s$(*state*, *abstractionsStates*[*state*].*representative*)
51    $d_s$(*state*, *new_abs.representative*) = get_distance(*state*, *new_abs.representative*)
52    **if** *new_abs* $\neq$ *abstractionsStates*[*state*] **or** $d_s$(*state*, *new_abs.representative*) $\neq$ *old_dist* **then**
53       **if** *state* == *abstractionsStates*[*state*].*representative* **and** *abstractionsStates*[*state*].*size* > 1 **then**
54          choose_new_representative_randomly(*abstractionsStates*[*state*], excluding *state*)
55          recalculate affected $d_s$ values
56       *abstractionsStates*[*state*] = *new_abs*
57       update_Q_abstraction(*state.parents*)

58 **function** get_distance($s_1, s_2$)
59    **if** $s_1$.*abstractActions* $\neq$ $s_2$.*abstractActions* **then**
60       **return** $\infty$
61    **foreach** *absQ* $\in$ $s_1$.*abstractActions* **do**
62       **if** $\left| \{ d_a(Q, absQ.representative) \mid Q \in absQ \cap s_1.actions \} \right|$ > 1 **or**
         $\left| \{ d_a(Q, absQ.representative) \mid Q \in absQ \cap s_2.actions \} \right|$ > 1 **then**
63          **return** $\infty$
64    *distances* = $\emptyset$
65    **foreach** *absQ* $\in$ $s_1$.*abstractActions* **do**
66       *distances.insert*$\big( d_s(p_1, p_2) \big)$          // $p_1, p_2 \in absQ,\ p_1 \in s_1.actions,\ p_2 \in s_2.actions$
67    **if** *distances.size* > 1 **then return** $\infty$ **else return** $d \in distances = \{d\}$
68

---

## A.2 PROOF THAT $d_a^* = d_a$ AND $d_s^* = d_s$

In this section, it will be shown that the KVDA difference functions $d_a$ and $d_s$ at any step (including the step of convergence) coincide with $d_a^*$ and $d_s^*$ for state or state-action pairs within the same abstract state. This will be proven inductively. First, note that this statement is true for the base case as all terminal states have a value of $V^* = 0$; hence, their differences are also 0. For the following induction steps, assume that $d_a$ is bootstrapped by $d_s'$ and $d_s$ is bootstrapped by $d_a'$. Next, let $(p_1, p_2) \in \mathcal{H}$. By definition, it holds that

$$d^*(p_1, p_2) = R(p_2) - R(p_1) + \underbrace{\sum_{s \in S} (\mathbb{P}(s \mid p_2) - \mathbb{P}(s \mid p_1)) V^*(s)}_{L:=} . \tag{7}$$

By rewriting $V^*(s)$ in terms of the difference to its abstract representative, one obtains

$$
\begin{aligned}
L &= \sum_{x \in \mathcal{X}} \sum_{s' \in x} (\mathbb{P}(s' \mid p_2) - \mathbb{P}(s' \mid p_1)) (V^*(s_x) - \underbrace{d_s^*(s', s_x)}_{=d_s' \text{ by induction}}) \\
&= \sum_{x \in \mathcal{X}} \sum_{s' \in x} (\mathbb{P}(s' \mid p_2) - \mathbb{P}(s' \mid p_1)) V^*(s_x) \\
&\quad + \sum_{x \in \mathcal{X}} \sum_{s' \in x} (\mathbb{P}(s' \mid p_1) - \mathbb{P}(s' \mid p_2)) \cdot d_s^*(s', s_x) \\
&= \sum_{x \in \mathcal{X}} V^*(s_x) \underbrace{\sum_{s' \in x} (\mathbb{P}(s' \mid p_2) - \mathbb{P}(s' \mid p_1))}_{=0} \\
&\quad + \sum_{x \in \mathcal{X}} \sum_{s' \in x} (\mathbb{P}(s' \mid p_1) - \mathbb{P}(s' \mid p_2)) \cdot d_s'(s', s_x) \\
&= d_a(p_1, p_2) + R(p_1) - R(p_2).
\end{aligned} \tag{8}
$$

Hence $d^*(p_1, p_2) = d_a(p_1, p_2)$. Note that this proof holds for any choice of the abstract nodes' representatives. Lastly, let $(s_1, s_2) \in \mathcal{E}$ and $d_s(s_1, s_2) = d$. First note that if $a \in \arg\max_{a' \in \mathbb{A}(s_1)} Q^*(s_1, a')$ and $((s_1, a), (s_2, \hat{a})) \in \mathcal{H}'$ with $d_a^*((s_1, a), (s_2, \hat{a})) = d$ for some $\hat{a} \in \mathbb{A}(s_2)$ then $\hat{a} \in \arg\max_{a' \in \mathbb{A}(s_2)} Q^*(s_2, a')$ because for all $b \in \mathbb{A}(s_2)$ it holds that

$$
\begin{aligned}
Q^*(s_2, \hat{a}) &= Q^*(s_1, a) - d_a^*((s_2, \hat{a}), (s_1, a)) \\
&\geq Q^*(s_1, \hat{b}) - d_a^*((s_2, \hat{a}), (s_1, a)) \\
&= Q^*(s_2, b) - d_a^*((s_2, \hat{b}), (s_1, b)) - d_a^*((s_2, \hat{a}), (s_1, a)). \\
&= Q^*(s_2, b) - (-d) + d = Q^*(s_2, b),
\end{aligned} \tag{9}
$$

where $((s_1, \hat{b}), (s_2, b)) \in \mathcal{H}'$ with $d_a^*((s_1, \hat{b}), (s_2, b)) = d$. And since $V^*(s) = \max_{a \in \mathbb{A}(s)} Q^*(s, a)$ it directly follows that $d = d_s(s_1, s_2) = d_s^*(s_1, s_2)$                    $\square$.

## A.3 RUNTIME MEASUREMENTS

Tab. 3 lists the average decision-making times for each environment of KVDA-UCT compared to OGA-UCT for 100 and 2000 iterations on states sampled from a distribution induced by random walks. The numbers show a median runtime overhead of $< 1\%$ for 100 iterations and $\approx 1\%$ for 2000 iterations.

Table 3: Average decision-making times in milliseconds of KVDA-UCT compared to OGA-UCT This data was obtained using an Intel(R) Core(TM) i5-9600K CPU @ 3.70GHz.

| Domain | KVDA-100 | OGA-100 | KVDA-2000 | OGA-2000 |
|---|---|---|---|---|
| Academic Advising | 3.85 | 3.82 | 133.82 | 131.74 |
| Cooperative Recon | 5.26 | 5.23 | 221.98 | 220.19 |
| Earth Observation | 7.54 | 7.54 | 197.38 | 199.13 |
| Elevators | 6.85 | 6.79 | 231.93 | 227.69 |
| Game of Life | 4.99 | 4.96 | 139.38 | 134.06 |
| Manufacturer | 10.26 | 11.01 | 281.03 | 294.91 |
| Red Finned Blue Eye | 6.69 | 6.66 | 152.46 | 151.56 |
| Sailing Wind | 4.11 | 4.09 | 139.39 | 140.82 |
| Saving | 3.75 | 3.71 | 124.22 | 111.13 |
| Skills Teaching | 4.97 | 4.94 | 219.90 | 216.68 |
| SysAdmin | 6.27 | 6.20 | 150.19 | 152.72 |
| Tamarisk | 4.65 | 4.64 | 167.23 | 148.03 |
| Traffic | 5.44 | 5.42 | 161.33 | 159.35 |
| Push Your Luck | 5.84 | 5.78 | 170.25 | 175.19 |
| Wildfire | 7.46 | 7.43 | 368.66 | 476.95 |
| Wildlife Preserve | 7.06 | 7.05 | 2020.35 | 1685.53 |
| Othello | 18.63 | 18.67 | 434.88 | 428.34 |
| Connect4 | 4.40 | 4.33 | 133.35 | 123.58 |
| Constrictor | 23.23 | 23.10 | 616.38 | 611.72 |

## A.4   DOMAIN-SPECIFIC $\varepsilon_a$ VALUES

One problem with $(\varepsilon_a, \varepsilon_t)$-OGA is that while $\varepsilon_t$ is neatly bounded by 0 and 2, the $\varepsilon_a$ value has to be chosen on a per-environment basis since for example the value $\varepsilon_a = 0.5$ would be equivalent to $\varepsilon_a = 0$ for any environment with rewards that are all greater than 0.5. Tab. 4 lists the hand picked values for each environment that we chose for the experiment section. The values were chosen to be at the boundary of rewards that actually occur in these environments.

Table 4: A list of the environment-specific $\varepsilon_a$ values that were evaluated in the experiment section for $(\varepsilon_a, 0)$-OGA. All domains that are not explicitly listed here, used the default values $\varepsilon_a \in \{1, 2, \infty\}$.

| Environment | $\varepsilon_a$ values |
|---|---|
| Academic Advising | $\infty$ |
| Wildlife Preserve | 5, 10, 20, $\infty$ |
| Red Finned Blue Eye | 50, 100, 200, $\infty$ |
| Wildfire | 5, 10, 100, $\infty$ |
| Push Your Luck | 2, 5, 10, $\infty$ |
| Skill Teaching | 2, 3, $\infty$ |
| Tamarisk | 0.5, 1.0, $\infty$ |
| Cooperative Recon | 0.5, 1.0, $\infty$ |
| Manufacturer | 10, 20, $\infty$ |
| Connect 4 | 1, 5, 10, $\infty$ |
| Othello | 5, 10, 20, $\infty$ |
| Constrictor | 10, 20, 30, $\infty$ |
| Default | 1, 2, $\infty$ |

## A.5 PERFORMANCES OF $\varepsilon_t$-KVDA

Table 5: The parameter-optimized performances of $\varepsilon_t$-KVDA (our method) and $(\varepsilon_a, \varepsilon_t)$-OGA on the stochastic versions of the here-presented environments using 1000 MCTS iterations (left) and 100 MCTS iterations (right). The environments Push Your Luck and Wildlife Preserve from the deterministic experiments were excluded as computing their transition probabilities which is a requirement for both $\varepsilon_t$-KVDA and $(\varepsilon_a, \varepsilon_t)$-OGA is infeasible. Furthermore, Red Finned Blue Eye, Wildfire and Elevators were also excluded as the performances' variances were so high in these settings such that obtaining low confidence bounds was infeasible. Contrary to the stochastic setting, $\varepsilon_t$-KVDA does not consistently perform better than OGA. For example, while $\varepsilon_t$-KVDA can gain a clear advantage in Manufacturer and Sailing Wind, $(\varepsilon_a, \varepsilon_t)$-OGA is significantly better in Tamarisk.

Average returns ($\uparrow$) for 1000 MCTS iterations

| Domain | $(\varepsilon_a, \varepsilon_t)$-OGA | $\varepsilon_t$-**KVDA** (ours) |
|---|---|---|
| s-Academic Advising | $-63.9 \pm 0.8$ | $\mathbf{-63.9 \pm 0.8}$ |
| s-Cooperative Recon | $\mathbf{14.3 \pm 0.2}$ | $14.1 \pm 0.2$ |
| s-Earth Observation | $-7.97 \pm 0.22$ | $\mathbf{-7.95 \pm 0.23}$ |
| s-Game of Life | $572.9 \pm 2.3$ | $\mathbf{573.4 \pm 2.2}$ |
| s-Manufacturer | $-1141.0 \pm 11.1$ | $\mathbf{-863.3 \pm 10.7}$ |
| s-Sailing Wind | $-62.1 \pm 1.3$ | $\mathbf{-61.8 \pm 1.3}$ |
| s-Saving | $\mathbf{50.7 \pm 0.1}$ | $50.6 \pm 0.1$ |
| s-Skills Teaching | $71.1 \pm 7.8$ | $\mathbf{74.9 \pm 6.9}$ |
| s-SysAdmin | $\mathbf{403.3 \pm 2.0}$ | $403.1 \pm 2.1$ |
| s-Tamarisk | $\mathbf{-532.9 \pm 8.2}$ | $-563.6 \pm 8.3$ |
| s-Traffic | $\mathbf{-13.4 \pm 0.3}$ | $-13.6 \pm 0.3$ |

Average returns ($\uparrow$) for 100 MCTS iterations

| Domain | $(\varepsilon_a, \varepsilon_t)$-OGA | $\varepsilon_t$-**KVDA** (ours) |
|---|---|---|
| s-Academic Advising | $-88.0 \pm 1.2$ | $\mathbf{-87.8 \pm 1.2}$ |
| s-Cooperative Recon | $\mathbf{7.08 \pm 0.37}$ | $7.07 \pm 0.35$ |
| s-Earth Observation | $\mathbf{-13.7 \pm 0.3}$ | $-13.8 \pm 0.3$ |
| s-Game of Life | $\mathbf{498.7 \pm 3.2}$ | $497.6 \pm 3.0$ |
| s-Manufacturer | $-1457.6 \pm 21.5$ | $\mathbf{-1159.6 \pm 24.8}$ |
| s-Sailing Wind | $-78.3 \pm 1.1$ | $\mathbf{-73.2 \pm 1.2}$ |
| s-Saving | $\mathbf{44.6 \pm 0.2}$ | $44.5 \pm 0.2$ |
| s-Skills Teaching | $\mathbf{-8.94 \pm 8.37}$ | $-9.60 \pm 8.23$ |
| s-SysAdmin | $\mathbf{332.7 \pm 2.8}$ | $331.1 \pm 2.7$ |
| s-Tamarisk | $\mathbf{-767.9 \pm 9.8}$ | $-836.1 \pm 7.8$ |
| s-Traffic | $\mathbf{-22.0 \pm 0.4}$ | $\mathbf{-22.0 \pm 0.4}$ |

## A.6 PERFORMANCES OF PARAMETER-OPTIMIZED KVDA-UCT FOR 200 AND 500 ITERATIONS

Table 6: Average returns ($\uparrow$) using 500 MCTS iterations for KVDA-UCT, $(\infty, 0)$-OGA, $(0, 0)$-OGA (i.e. OGA-UCT), and the maximal performance of $(\varepsilon_a, 0)$-OGA when varying $\varepsilon_a > 0$. In the supplementary materials in Tab.4 the domain-specific $\varepsilon_a$-values are listed. All performances are the maximal performances obtained by varying the exploration constants $C \in \{0.5, 1, 2, 4, 8, 16\}$. Note that KVDA-UCT (our method) outperforms OGA-UCT in most environments. Furthermore, KVDA-UCT is either equal or performs better than parameter-optimized $(\varepsilon_a, 0)$-OGA even though KVDA-UCT introduces no extra parameter.

| Domain | $(0, 0)$-OGA | $(\infty, 0)$-OGA | $(\varepsilon_a, 0)$-OGA | **KVDA-UCT** (ours) |
|---|---|---|---|---|
| Connect4 | $36.7 \pm 0.5$ | $\mathbf{41.3 \pm 1.0}$ | $\mathbf{41.3 \pm 1.0}$ | $39.9 \pm 0.6$ |
| Constrictor | $\mathbf{94.2 \pm 0.2}$ | $92.8 \pm 0.2$ | $94.2 \pm 0.3$ | $94.0 \pm 0.2$ |
| Othello | $\mathbf{175.3 \pm 1.1}$ | $156.8 \pm 0.8$ | $175.2 \pm 1.6$ | $173.4 \pm 1.1$ |
| Pusher | $104.5 \pm 0.0$ | $104.5 \pm 0.0$ | $104.5 \pm 0.0$ | $\mathbf{104.5 \pm 0.0}$ |
| d-Academic Advising | $-42.1 \pm 0.3$ | $\mathbf{-42.1 \pm 0.3}$ | $\mathbf{-42.1 \pm 0.3}$ | $-42.3 \pm 0.3$ |
| d-Cooperative Recon | $15.3 \pm 0.1$ | $14.4 \pm 0.1$ | $\mathbf{15.4 \pm 0.2}$ | $15.3 \pm 0.2$ |
| d-Earth Observation | $-7.28 \pm 0.12$ | $-29.6 \pm 0.4$ | $-29.6 \pm 0.4$ | $\mathbf{-6.98 \pm 0.10}$ |
| d-Elevators | $-15.6 \pm 0.3$ | $\mathbf{-14.6 \pm 0.3}$ | $\mathbf{-14.6 \pm 0.3}$ | $-14.9 \pm 0.3$ |
| d-Game of Life | $659.9 \pm 2.6$ | $659.9 \pm 2.6$ | $660.7 \pm 2.5$ | $\mathbf{661.6 \pm 2.5}$ |
| d-Manufacturer | $-1270.2 \pm 11.7$ | $-1646.1 \pm 20.8$ | $-1259.1 \pm 16.5$ | $\mathbf{-1178.3 \pm 22.3}$ |
| d-Push Your Luck | $124.2 \pm 1.9$ | $67.4 \pm 1.0$ | $131.4 \pm 2.0$ | $\mathbf{136.5 \pm 1.9}$ |
| d-RedFinnedBlueEye | $7978.4 \pm 32.9$ | $7738.1 \pm 34.2$ | $7777.4 \pm 34.3$ | $\mathbf{8041.7 \pm 33.1}$ |
| d-Sailing Wind | $-43.4 \pm 0.7$ | $-42.8 \pm 0.7$ | $-42.8 \pm 0.7$ | $\mathbf{-41.6 \pm 0.7}$ |
| d-Saving | $\mathbf{64.5 \pm 0.2}$ | $61.7 \pm 0.2$ | $64.2 \pm 0.2$ | $64.4 \pm 0.2$ |
| d-Skills Teaching | $202.8 \pm 3.7$ | $201.0 \pm 3.9$ | $202.6 \pm 3.9$ | $\mathbf{206.5 \pm 3.7}$ |
| d-SysAdmin | $477.7 \pm 1.5$ | $448.8 \pm 1.2$ | $450.8 \pm 1.2$ | $\mathbf{478.1 \pm 1.5}$ |
| d-Tamarisk | $-228.9 \pm 4.1$ | $-217.9 \pm 2.4$ | $-217.9 \pm 2.4$ | $\mathbf{-197.2 \pm 2.7}$ |
| d-Traffic | $-2.60 \pm 0.10$ | $\mathbf{-2.53 \pm 0.09}$ | $\mathbf{-2.53 \pm 0.09}$ | $\mathbf{-2.53 \pm 0.09}$ |
| d-Wildfire | $\mathbf{-183.6 \pm 36.1}$ | $-506.0 \pm 36.2$ | $-409.6 \pm 36.1$ | $-190.6 \pm 36.3$ |
| d-Wildlife Preserve | $1385.1 \pm 0.8$ | $1384.2 \pm 0.8$ | $\mathbf{1385.5 \pm 0.8}$ | $1385.0 \pm 0.8$ |

Table 7: Average returns using 200 ($\uparrow$) MCTS iterations for KVDA-UCT, $(\infty, 0)$-OGA, $(0, 0)$-OGA (i.e. OGA-UCT), and the maximal performance of $(\varepsilon_a, 0)$-OGA when varying $\varepsilon_a > 0$. In the supplementary materials in Tab.4 the domain-specific $\varepsilon_a$-values are listed. All performances are the maximal performances obtained by varying the exploration constants $C \in \{0.5, 1, 2, 4, 8, 16\}$. Note that KVDA-UCT (our method) outperforms OGA-UCT in most environments. Furthermore, KVDA-UCT is either equal or performs better than parameter-optimized $(\varepsilon_a, 0)$-OGA even though KVDA-UCT introduces no extra parameter.

| **Domain** | $(0, 0)$-OGA | $(\infty, 0)$-OGA | $(\varepsilon_a, 0)$-OGA | **KVDA-UCT** (ours) |
|---|---|---|---|---|
| Connect4 | $29.1 \pm 0.4$ | $\mathbf{29.9 \pm 0.6}$ | $\mathbf{29.9 \pm 0.6}$ | $29.5 \pm 0.4$ |
| Constrictor | $91.3 \pm 0.3$ | $90.2 \pm 0.2$ | $\mathbf{91.3 \pm 0.4}$ | $91.2 \pm 0.3$ |
| Othello | $164.5 \pm 1.2$ | $152.1 \pm 0.8$ | $\mathbf{164.7 \pm 1.7}$ | $164.5 \pm 1.2$ |
| Pusher | $104.5 \pm 0.0$ | $\mathbf{104.5 \pm 0.0}$ | $\mathbf{104.5 \pm 0.0}$ | $104.5 \pm 0.0$ |
| d-Academic Advising | $-47.3 \pm 0.4$ | $\mathbf{-47.2 \pm 0.4}$ | $\mathbf{-47.2 \pm 0.4}$ | $-47.5 \pm 0.4$ |
| d-Cooperative Recon | $\mathbf{13.2 \pm 0.3}$ | $12.7 \pm 0.3$ | $13.1 \pm 0.3$ | $13.1 \pm 0.3$ |
| d-Earth Observation | $-7.82 \pm 0.14$ | $-29.3 \pm 0.2$ | $-29.3 \pm 0.2$ | $\mathbf{-7.11 \pm 0.11}$ |
| d-Elevators | $-16.2 \pm 0.3$ | $-16.1 \pm 0.3$ | $-16.1 \pm 0.3$ | $\mathbf{-16.0 \pm 0.3}$ |
| d-Game of Life | $651.4 \pm 2.5$ | $\mathbf{652.3 \pm 2.4}$ | $652.3 \pm 2.4$ | $651.1 \pm 2.5$ |
| d-Manufacturer | $-1370.7 \pm 15.3$ | $-1641.4 \pm 21.2$ | $-1320.7 \pm 18.6$ | $\mathbf{-1180.8 \pm 22.2}$ |
| d-Push Your Luck | $122.6 \pm 1.9$ | $67.1 \pm 0.9$ | $\mathbf{129.6 \pm 1.9}$ | $125.9 \pm 1.8$ |
| d-RedFinnedBlueEye | $7713.6 \pm 33.9$ | $7511.8 \pm 34.7$ | $7513.4 \pm 34.9$ | $\mathbf{7716.9 \pm 33.7}$ |
| d-Sailing Wind | $-52.7 \pm 0.8$ | $-52.3 \pm 0.8$ | $-52.3 \pm 0.8$ | $\mathbf{-51.7 \pm 0.8}$ |
| d-Saving | $\mathbf{60.9 \pm 0.2}$ | $58.2 \pm 0.2$ | $60.5 \pm 0.2$ | $60.6 \pm 0.2$ |
| d-Skills Teaching | $183.9 \pm 3.9$ | $\mathbf{186.7 \pm 3.8}$ | $\mathbf{186.7 \pm 3.8}$ | $185.0 \pm 3.8$ |
| d-SysAdmin | $476.6 \pm 1.6$ | $450.2 \pm 1.2$ | $450.3 \pm 1.2$ | $\mathbf{478.9 \pm 1.4}$ |
| d-Tamarisk | $-265.1 \pm 5.5$ | $-242.9 \pm 3.5$ | $-242.9 \pm 3.5$ | $\mathbf{-221.7 \pm 3.8}$ |
| d-Traffic | $\mathbf{-5.34 \pm 0.15}$ | $\mathbf{-5.34 \pm 0.15}$ | $\mathbf{-5.34 \pm 0.15}$ | $-5.34 \pm 0.15$ |
| d-Wildfire | $-193.8 \pm 36.1$ | $-476.5 \pm 36.1$ | $-432.2 \pm 36.2$ | $\mathbf{-191.2 \pm 36.1}$ |
| d-Wildlife Preserve | $\mathbf{1390.1 \pm 0.9}$ | $1389.4 \pm 0.9$ | $1389.4 \pm 0.9$ | $1389.5 \pm 1.0$ |

## A.7 PERFORMANCE GRAPHS OF PARAMETER-OPTIMIZED KVDA-UCT

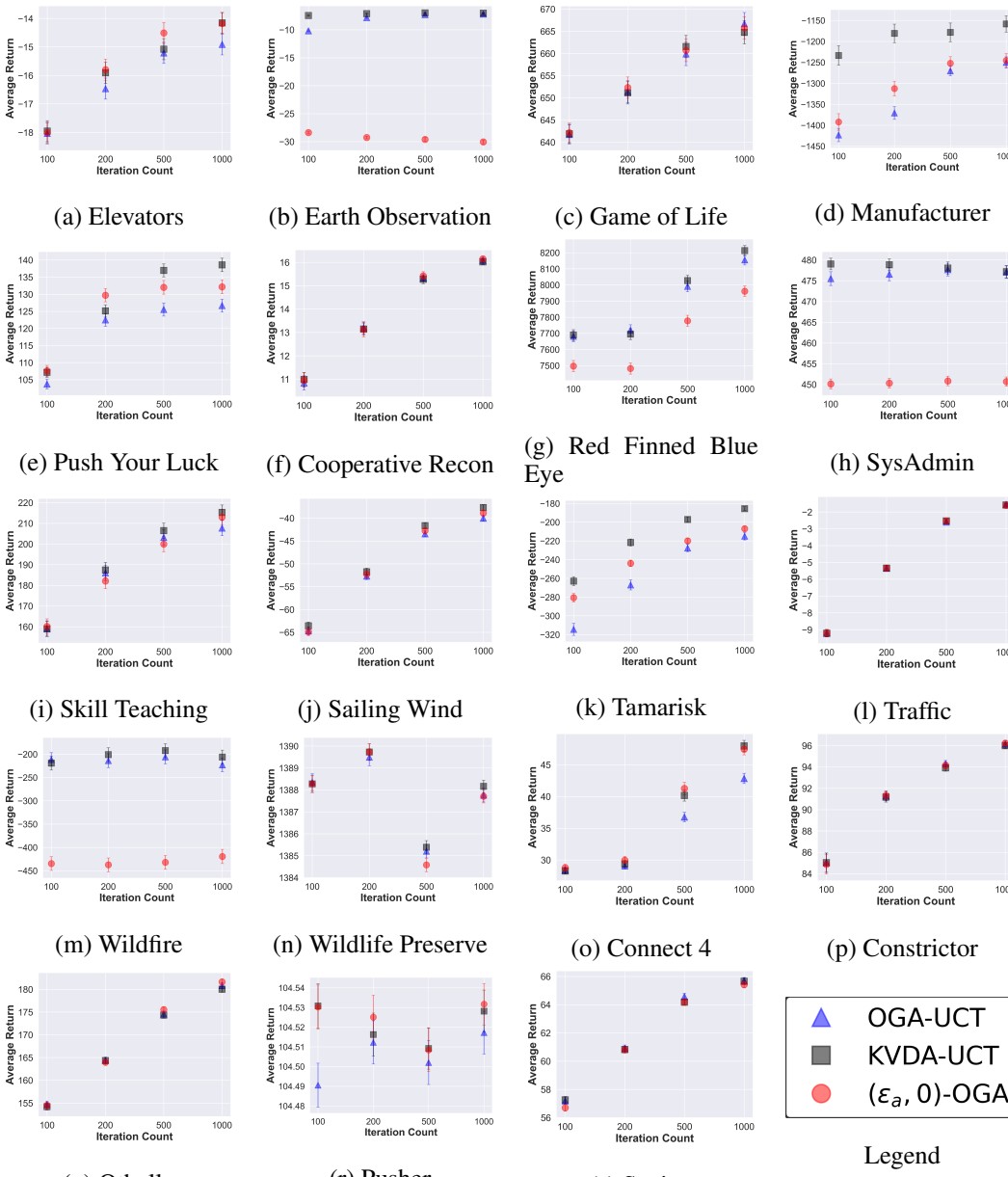

Figure 4: The performance plots in dependence of the iteration budget of parameter-optimized KVDA-UCT (our method), OGA-UCT, and $(\varepsilon_a, 0)$-OGA, $\varepsilon_a > 0$ on all considered environments. Our method KVDA-UCT is either always tightly within the confidence bounds of the top-competitor method or outperforms all competitors simultaneously.

## A.8 MONTE CARLO TREE SEARCH

This section specifies the MCTS that was used for the experiments of this paper.

1. Our MCTS version builds a directed acyclic graph instead of a tree, i.e. different state-action pairs in the same layer can lead to the same node. This is a necessary condition for ASAP and KVDA to detect any abstractions since they bootstrap of the search graph converging.

2. We use the Upper Confidence Bounds (UCB) tree policy which is defined as

$$\text{UCB}(a) = \underbrace{\frac{V_a}{N_a}}_{\text{Q term}} + \underbrace{\lambda\sqrt{\frac{\log\left(\sum_{a'\in\mathbb{A}(s)} N_{a'}\right)}{N_a}}}_{\text{Exploration term}}, \tag{10}$$

where, $s$ is the state at which the decision has to be made, $V_a$ is the sum of returns of the action under consideration, and $N_a$ are its visits. The tree policy chooses the action that maximizes the UCB value.

3. We use the greedy decision policy, which is choosing the root action, after the search statistics have been gathered, with the maximum Q value.

### A.9 PROBLEM DESCRIPTIONS

The descriptions for all the problem models considered here are described in detail by Schmöcker et al. Schmöcker et al. (2025b). Most models are parametrized (e.g. choosing a concrete racetrack for Racetrack), the concrete instance used for the experiments is found in the *ExperimentConfigs* folder in our publicly available implementation (Schmöcker, 2025). Next, we describe the heuristic functions used for the two-player games.

1. **Connect 4**: As the heuristic from the perspective of player one is given by

$$n_2 + 5n_3 + 25n_4 \tag{11}$$

where $n_i$ is the total number of $i$ stones that are in one row/column/diagonal, divided by $i$, but which are not part of a row/column/diagonal of size $i + 1$.

2. **Constrictor**: The heuristics function used for player one is the number of grid cells that player one could reach before its opponent. If player one wins, the value $100$ is added to the heuristic.

3. **Othello**: The heuristic function for any state for player one is given by a weighted sum of all the occupied grid cells. The weight $w$ of cell $(x, y)$ is given by $10/(1 + d)$ where $d$ is the distance to the closest corner cell. The weight $w$ is positive for any cells occupied by stones of player one and negative for those occupied by player two. In the round player one wins, $100$ is added to the heuristics value and $-100$ if player two won.

4. **Pusher**: We used the heuristic for player one that is equal to the difference of alive units of player one and player two in addition to the Manhatten distance between player one and player two's units' center of mass. In the turn that player one wins, $100$ is added to the heuristic value.

