# OpenReview forum: "Grouping Nodes with known Value Differences: A lossless UCT-based Abstraction Algorithm"
_ICLR.cc/2026/Conference — ICLR 2026 Poster_

### Official Review · Reviewer_EN6Y · 2025-11-01

**Soundness:** 2
**Presentation:** 2
**Contribution:** 2
**Rating:** 4
**Confidence:** 3

**Summary:**

This paper proposes Known Value Difference Abstractions (KVDA), a framework that extends the ASAP (Abstractions of State-Action Pairs) methodology for Monte Carlo Tree Search (MCTS) in deterministic environments. Unlike ASAP which groups only state-action pairs with identical Q-star values, KVDA deliberately groups pairs with different values if their difference can be inferred from immediate rewards. The authors implement KVDA-UCT (a modification of OGA-UCT) that tracks difference-adjusted values during aggregation and a generalized version (epsilon_t-KVDA) for stochastic settings. Experimental evaluation across 19 deterministic domains and their stochastic variants shows KVDA-UCT detects substantially more abstractions than OGA-UCT and typically matches or exceeds parameter-optimized (epsilon_a, 0)-OGA performance without introducing additional parameters.

**Strengths:**

Elegant and Sound Conceptual Contribution: The core insight relaxing value-equivalence to known value differences is a natural and well-motivated extension of ASAP. Figure 1 effectively demonstrates cases where ASAP fails but KVDA succeeds. The difference-accounted aggregation mechanism is theoretically sound, and the proof correctly establishes the exactness (lossless) guarantee.

Strong Empirical Performance (Deterministic): The method is practical, as KVDA-UCT introduces no new parameters. It empirically detects significantly more abstractions than OGA-UCT. Crucially, it consistently matches or outperforms a parameter-optimized (epsilon_a, 0)-OGA in both raw performance and generalization, all while maintaining negligible runtime overhead.

Comprehensive Methodology and Reproducibility: The evaluation is thorough, covering multiple metrics (abstraction detection, optimized performance, generalization), various iteration budgets, and detailed domain descriptions. The authors also provide reproducible code with compilation details.

**Weaknesses:**

Poor Stochastic Performance: The method's advantages largely disappear in the approximate (stochastic) setting. epsilon-t-KVDA exhibits mediocre performance, sometimes significantly underperforming. The authors acknowledge this stems from "faulty abstractions" but offer it only as future work, substantially limiting the method's practical applicability.

Experimental Design Lacks Critical Controls: The use of hand-engineered heuristics to create dense rewards for board games may artificially favor KVDA's reward-difference mechanism. Furthermore, some domains show identical performance across all methods and some comparisons have overlapping confidence intervals, yet are still presented as favorable to KVDA.

Significantly Restricted Scope: The paper explicitly states the method does not extend to multi-player games due to the lack of a unique V-star. It also appears dependent on deterministic immediate rewards (not addressing stochastic rewards). These constraints sharply limit the method's impact, as MCTS is highly prevalent in the exact domains that are excluded.

**Questions:**

Stochastic Failure Modes: The paper attributes epsilon-t-KVDA's poor performance to faulty abstractions. Can you precisely characterize what type of faulty abstractions occur, and why the aggregation/representative-switching mechanism fails to correct for them under transition errors?

Impact of Heuristics: The board game results rely on dense reward transformations via hand-engineered heuristics. How robust are the performance gains to different heuristic choices?

Multi-Player Extension: The paper notes V-star is not unique in multi-player settings. Could the KVDA framework be adapted to work with different value concepts, such as Nash equilibrium values or empirical best-response values from self-play, to broaden its applicability?

---

> ### Author Response · Authors · 2025-11-18
> **Answers**
>
> R3.1: "Poor Stochastic Performance..."
>
> A3.1: Yes, you are right, but like multi-player games, stochastic environments were not the focus of this method. The aim was to fill a gap in the literature since most abstraction techniques only work to improve ASAP in stochastic settings (e.g. EMCTS, intra-abstraction policies, or pruned OGA). We added the following paragraph to the introduction for better positioning our work:
>     "The introduction of KVDA-UCT..." (see revised Paper)
>
>     - Furthermore, we'd like to point out that one can also view the stochastic results from an optimistic standpoint. KVDA performs only clearly worse in Tamarisk, while clearly beating OGA in Manufacturer and Sailing Wind for 100 iterations. Hence, one can also see KVDA for stochastic environments as another tool for one's toolbox when trying to optimize for performance.
>
>
> R3.2 and R3.3: "Experimental Design Lacks Critical Controls..." and "Impact of Heuristics..."
>
> A3.2 and A3.3: Perhaps, yes but note that these environments mostly do not feature any performance gains by KVDA which are at the risk of being falsely advertised. Furthermore, the results on these heuristics do not vary qualitatively from those obtained at the other MDP environments which do not use any heuristics.
>
>
> R3.4: "Furthermore, some domains show identical performance across all methods"
>
> A3.4: We acknowledge that our method can not find a performance improvement in every domain. However, that is to be expected since not every domain features value equivalent states or state-action pairs that can be grouped by KVDA but not by ASAP.
>
>
> R3.5: "and some comparisons have overlapping..."
>
> A3.5: While carefully reviewing our work, we sadly could not spot the referenced text-passage. Can you say to which line this refers?
>
> R3.6: "Significantly Restricted Scope..."
>
> A3.6:
>     - Multi-agent settings: While we agree that MCTS is highly prelevant in multi-agent domains, it is certainly also widely used in stochastic MDP settings as those examined in this paper (see the International Conference in Probabilistic Planning for example).
>     - Deterministic immediate rewards: Though we only tested the method for deterministic immediate rewards, the method is in principle still applicable to stochastic reward settings by using samples the estimate the average immediate reward. Therefore this does not restrict the scope. We conducted no experiments in this setting since the literature on abstractions for MCTS (the ones cited in the paper) exclusively use deterministic immediate rewards.
>
>
> R3.7: "Multi-Player Extension..."
>
> A3.7: We have thought hard about this problem and could not find a solution. The key problem is that there can, in general, be multiple Nash Equilibria with different payoff vectors. Therefore, the value function differences d_a and d_s become ill-defined.  Could you please explain what you mean by "empirical best-response" and how that is related to the MCTS version used here. So far we explicitly mention this 'weakness' and leave it as future work.
>
>
> R3.8: "Stochastic Failure Modes..."
>
> A3.8: When choosing an epsilon_t value > 0, then both ASAP and KVDA could group state-action pairs with different Q* values even if the search tree below is fully expanded. The problem that occurs with KVDA is that is amplifies the number of faulty abstractions that are bootstrapped of such a state-action pair. Concretely, the following table shows the performances for the Tamarisk environment, where eps_t-KVDA is outperformed by OGA. Clearly, eps_t-KVDA performance degrades much faster with an increase abstraction coarseness. The following table show the performances for 1k iterations:
>
> | eps_t |   oga   |   kvda   |
> |-------|---------|----------|
> | 0.0   | -558  | -561   |
> | 0.2   | -548  | -578   |
> | 0.4   | -537  | -619   |
> | 0.8   | -542  | -872   |
> | 1.2   | -597  | -1098  |
> | 1.6   | -754  | -1252  |
>
>     - For transparency purposes, however, we like to state that we cannot definitively prove that this is the mechanism at play here. Therefore, we wrote "we believe" in the paper.
>
> Further remarks: Though you did not criticize this, but the empirical results are actually better than presented in the initial paper, simply due to the fact that we accidentally misplaced the column names in Table 2 for 1000 iterations. The KVDA column has to be swapped with the (e_a,0)-OGA column. This mistake has been spotted by another reviewer and we just want to mention this change for completeness. We highly encourage you to revisit Table 2 since this immensely improves the significance of KVDA-UCT.
>
> We also rewrote some sections based on the feedback received to improve the wording and flow of text. The following sections are affected.
>     -slight pruning of related work to improve flow of text
>     -added paragraph on differences regarding bisimulation approaches
>     -slight pruning of mcts + environment descriptions in appendix
>     -fixed spelling/grammar mistakes + some rewording

---

### Official Review · Reviewer_iyK8 · 2025-11-01

**Soundness:** 3
**Presentation:** 2
**Contribution:** 1
**Rating:** 4
**Confidence:** 4

**Summary:**

Authors propose KVDA, a novel MCTS lossless abstraction method rooted from ASAP. KVDA modifies ASAP’s abstraction criteria, especially removing the criterion of immediate reward equivalence, and leaving the Q differences between the Q nodes as a record when abstracted. For state nodes, KVDA abstracts two state nodes if their all Q nodes are abstracted to each other and Q differences of their all abstractions are same. Authors merge KVDA with UCT as KVDA-UCT, then compare abstraction ratio and general performance of KVDA-UCT with OGA-UCTs on various board games, proving that KVDA-UCT leads more abstractions than OGA-UCT, and that KVDA-UCT outperforms OGA-UCT mainly on deterministic environments.

**Strengths:**

The idea is distinct and sound, supported with theoretical guarantees. Background description is elaborate and well grounded. Authors adopt adequate counterparts for comparison experiments, which are conducted on appropriate domain and various tasks. They also consider extending KVDA to stochastic environments for generality.

**Weaknesses:**

#### Major Weaknesses

- KVDA–OGA Performance Ambiguity: The biggest concern is that according to Table 2, different from the authors’ comment, the performance of (ε_a, 0)-OGA with 1,000 iterations seems to generally outperform the KVDA. If it’s the case, authors might claim that since KVDA more generously abstracts than (ε_a, 0)-OGA, KVDA aggressively finds good paths more efficiently under the condition of low iteration budget. Here, at least I can agree that KVDA is better than (∞, 0)-OGA, since (∞, 0)-OGA would yield lots of faulty abstractions while KVDA would not because it’s lossless. This is where KVDA mitigates the potential negative effects of its generosity and turns it to its advantage. Still, if (ε_a, 0)-OGA performs poorly in lower iterations but achieves the best performance at 1,000 iterations, it may imply that the conservative aspect of OGAs contributes to steady but ultimately powerful performance growth. Since 1,000 iterations are not practically high budget, if this trend reproduced in higher iterations, the contribution of KVDA would become more limited, unless OGAs only outperform in practically very high budget. Because KVDA struggled in stochastic envs (and I agree leaving improvement of it for future work), proving the effects of KVDA in deterministic envs would naturally be emphasized to highlight the sufficient significance of the paper.
- Insufficient Method Discussion: Secondly, the lack of discussion on the main method makes authors’ suggestions less convincing. Throughout the paper, it appears relatively limited compared to other parts such as background descriptions and equations. Although introducing the foundations and ASAP (including subsection 3.1) is necessary (But are Sections 2.3 and 2.4 really necessary as standalone subsections?), it would be more convincing and provide denser insight to focus more on the main proposal, especially by providing theoretical or empirical analysis (e.g., a few actual abstraction examples obtained from the experiments) of the expected positive/negative effects due to the KVDA’s lossless but generous abstraction conditions. Explanation of why the results were so good for some tasks and vice versa might also help.
- Poor Structural Cohesion: Lastly, the overall flow of the paper is not very cohesive and somewhat difficult to follow. For instance, it was hard to grasp the key concept of main idea only reading Introduction. The initial introduction of understandable mechanism of the main method is introduced somewhat too late, which made reading before subsection 3.2 a little exhausting. I cautiously suggest adopting a “method-first, explanation-later” structure for the Method section, or providing brief mechanical information about KVDA in the preceding sections. Some sentences were too lengthy or colloquial. Figures and Tables of experimental results were confusing due to their jumpy locations and several formatting errors. Experimental conditions for each result were also confusing (especially the differences between Table 2, 6, 7 and Figure 4. Why do they appear to show different results?). Refining the overall flow with more intuitive and concise writing would be needed.

#### Minor Weaknesses
- Equation 6, it appears that the sign of d was not considered.
- Page 7, unnecessary three “Section 5”?
- Page 13, 14, Table errors
- Page 26, Image error
- Reproducibility link is unavailable.
- Figure 3 (Page 27), numbering error
- Figure 4 (Page 16), legend “(ε_a, εt)-OGA” instead of “(ε_a, 0)-OGA”?

**Questions:**

All questions and suggestions are included in the Weaknesses.

---

> ### Author Response · Authors · 2025-11-18
> **Answers**
>
> In the following, we marked points by the reviewer with R2.x and our corresponding answers with A2.x:
>
> R2.1: "KVDA–OGA Performance Ambiguity..."
>
> A2.1: You are absolutely right to point this out and we did a huge blunder here. Actually the fourth and last column for the 1000 iterations setting must be swapped. I.e. the (e_a,0)-OGA column is actually the KVDA column. And as you rightly pointed out this column, which is actually KVDA, performs best overall. This has been updated in the revised version of the paper. We are sorry for this mistake and the confusion that has been caused by it. Having addressed this issue, we also ensured that all the other reported experimental results are correctly displayed in the paper.
>
>
> R2.2: "Insufficient Method Discussion..."
>
> A2.2: Firstly to address your concern to improve our method's description, we added a detailed pseudocode in the appendix and modified the section introduction as follows:
>
>      "This section introduces our novel KVDA-UCT algorithm which is designed to detect strictly more abstractions than OGA-UCT which allows one to compute more accurate UCB values during both the decision and tree policy which in turn leads to performance improvements."
>
> Furthermore, we updated items 3) and 5) in Section 3.3:
>
>     3) "For efficient distance function calculations, each abstract node keeps track of a representative. Value differences are only calculated with respect to the representative. An abstract node's representative is the first original node added to the abstract node, and if that one is removed, a random new representative is chosen."
>
>     5) "Since the KVDA abstraction also depends on the difference functions, whenever a Q-node's recency counter reaches the threshold, the value difference to its representative is recalculated. A change in this difference also results in a reevaluation (and subsequent recency counter reset) of the parent nodes' abstractions and difference functions. "
>
>     - Next, to provide more insights into the concrete abstractions that KVDA can find which ASAP cannot, we added the following paragraph to Section 3.1:
>
>     "While this was only a toy... (see revised Paper)"
>
>
> R2.3: "But are Sections 2.3 and..."
>
> A2.3: For further clarification, by 'are Sections 2.3 and 2.4 really necessary as standalone subsections?' do you mean merging them into one subsection or simply removing them? We agree that they can be merged but would refrain from removing them due to positive comments on the background provided in Section 2 meant to provide an overview of required preliminaries for the remainder of the article.
>
>
> R2.4: "Explanation of why ..."
>
> A2.4: Since we simply mislabelled the columns, there is no longer a single deterministic environment which KVDA is worse than OGA. We believe that this addresses your remark 'Explanation of why the results were so good for some tasks and vice versa might also help.'
>
>
> R2.5: "Poor Structural Cohesion..."
>
> A2.5: Could you guide us here by sharing which misconception or unanswered questions you had while reading the introduction? We can then insert this missing piece of information
>
>
> R2.6: "The initial introduction of understandable mechanism..."
>
> A2.6: Thank you for the suggestion. A method-first approach was our intent. In fact, subsection 3.1. was supposed to act as a high-level KVDA explanation. To make this more clear, we added the following paragraph to section 3.1:
>
>     "The analysis of this example... (see revised Paper)"
>
> R2.7: "Some sentences were too lengthy or colloquial..."
>
> A2.7: Thanks for the detailed list of required changes. Again we are sorry for the initial mistake regarding the presentation of our data. We fixed all the formatting issues and moved Figures 2 and 3 directly into the main part. Furthermore, as mentioned above the experimental results should no longer be confusing given that we simply mislabelled the two columns in Table 2.
>
>
> R2.8:
> "Equation 6..."
> "Page 7..."
> "Page 13, 14..."
> "Page 26..."
> "Figure 3 ..."
> "Figure 4..."
>
> A2.8 Thanks, for marking these errors. We fixed them in the updated version of our paper.
>
>
> R2.9: "Reproducibility link..."
>
> A2.9: In some cases, the PDF-link does not forward to the correct web-address due to special characters in the URL. Please directly enter the following website into your browser and confirm if the repository is acessible:
>
> 	https://anonymous.4open.science/r/KVDA_UCT-6E51/README.md
>
> For the final paper, we will ensure that the original github repository is made public and the appropriate link is referenced in the paper.
>
> Additionally, we rewrote some sections based on the feedback received to improve the wording and flow of text. The following sections are affected.
>     - slight pruning of related work to improve flow of text
>     - added paragraph on differences regarding bisimulation approaches
>     - slight pruning of mcts and environment descriptions in appendix
>     - fixed spelling and grammar mistakes as well as some rewording

---

### Official Review · Reviewer_VwR2 · 2025-11-09

**Soundness:** 3
**Presentation:** 2
**Contribution:** 3
**Rating:** 6
**Confidence:** 2

**Summary:**

The paper tackles the problem of low sample efficiency in Monte Carlo Tree Search (MCTS). Existing methods, such as OGA-UCT, improve efficiency by abstracting similar states and state-action pairs, but rely on strict equality of immediate rewards, which limits the scope of abstraction. The authors propose a new framework, Known Value Difference Abstractions (KVDA), which allows grouping states or actions with different but inferable value differences. Incorporating KVDA into OGA-UCT (forming KVDA-UCT) leads to more abstractions, better efficiency, and improved performance across various deterministic environments, all without introducing extra parameters.

**Strengths:**

- The problem tackled by this work is well motivated. Increasing the number of abstractions helps improve the sample efficiency of MCTS.
- The solution, proposed in this work, sounds. The idea of abstraction using a known difference in values between state(-action) pairs is novel to me.
- I really appreciate Section 2, which is essential to understand the preliminaries and know the related work.
- I believe the results are strong and the proposed abstraction technique does not hurt in terms of performance, except in the stochastic case.
- The limitations are clearly mentioned.

**Weaknesses:**

- I believe some parts of the main paper need to be adjusted for clarity.
- There are some tables and figures that are placed in the Appendix while being discussed as main results in the main paper. I understand the issue with space, but I believe it is really important for the main paper to be isolated from the appendix.
- This is especially because the appendix is not well-presented, since there are some figures that are not inserted correctly.
- I believe the baselines benchmarked in this work are fair, but I would be interested in looking into some other methods that were already discussed and mentioned in Section 2.

**Minor Issues:**
- There is an issue with referencing the sections in lines 376-377.

**Questions:**

- In the deterministic setting, when might the proposed abstraction (based on the known difference in values) harm the performance?

---

> ### Author Response · Authors · 2025-11-18
> **Answers**
>
> In the following, we marked points by the reviewer with R1.x and our corresponding answers with A1.x:
>
> R1.1: "I believe some parts of the main paper need to be adjusted for clarity."
>
> A1.1: Could you point us towards a section/paragraph which you think was difficult to parse? We will be happy to revise the corresponding text sections to improve our paper.
>
>
> R1.2: "There are some tables and figures that are placed in the Appendix while being discussed as main results in the main paper. I understand the issue with space, but I believe it is really important for the main paper to be isolated from the appendix."
>
> A1.2: Thanks for noting this. We moved both Figure 3 (the performance scores) as well as the performance graphs (Figure 2) fromthe appendix into the main part.
>
>
> R1.3: "This is especially because the appendix is not well-presented, since there are some figures that are not inserted correctly."
>
> A1.3: We fixed the formatting of all Figures and Tables that appeared in the appendix.
>
>
> R1.4: "I believe the baselines benchmarked in this work are fair, but I would be interested in looking into some other methods that were already discussed and mentioned in Section 2."
>
> A1.4: Thanks for pointing this out, but we believe these do not fit our scope, since the focus of KVDA-UCT was to develop a method for deterministic settings. Intra-abstraction policies, abstraction dropping, or determinization approaches were all designed for stochastic environments to deal with highly lossy abstractions which is not the case here. We addressed this by refining our scope, hence we inserted the following in the introduction:
>
>     "The introduction of KVDA-UCT aims to address the abstraction literature gap for deterministic settings since existing modifications of OGA-UCT have only focused on stochastic settings (Anand et al., 2016; Xu et al., 2023; Schmöcker et al., 2025c;b)."
>
>
> R1.5: "There is an issue with referencing the sections in lines 376-377."
>
> A1.5: Thanks, this has been fixed in the updated version!
>
>
> Further remarks: Though you did not criticize this, but the empirical results are actually better than presented in the initial paper, simply due to the fact that we accidentally misplaced the column names in Table 2 for 1000 iterations. The KVDA column has to be swapped with the (e_a,0)-OGA column. This mistake has been spotted by another reviewer and we just want to mention this change for completeness. We highly encourage you to revisit Table 2 since this immensely improves the significance of KVDA-UCT.
>
> Additionally, we rewrote some sections based on the feedback received to improve the wording and flow of text. The following sections are affected.
>     - slight pruning of related work to improve flow of text
>     - added paragraph on differences regarding bisimulation approaches
>     - slight pruning of mcts and environment descriptions in appendix
>     - fixed spelling and grammar mistakes as well as some rewording
>
>
> Questions:
> - "In the deterministic setting, when might the proposed abstraction (based on the known difference in values) harm the performance?"
>     - The only setting which we can think of, is when two state-action pairs with different immediate rewards and different Q-star values have not fully sampled their successor states such that the incomplete successor sets are identical. Then these would be grouped by KVDA but not by ASAP.

---

### Author Response · Authors · 2025-11-19
**Difference highlighted version**

We uploaded the differences-highlighted revision as supplementary materials such that the reviewers can more easily see the changes made.

---

### Meta-Review · Area_Chair_8kF1 · 2026-01-14

**Summary:**

The paper proposes Known Value Difference Abstractions (KVDA), a framework extending ASAP (Abstractions of State-Action Pairs) for MCTS. Unlike previous methods that require strict value equivalence or identical immediate rewards, KVDA groups state-action pairs with known value differences inferred from immediate rewards. The method is proven to be lossless in deterministic environments. Reviewers generally appreciated the novelty of the difference-based abstraction (VwR2, EN6Y) and the theoretical soundness in the deterministic setting. Concerns were raised regarding the paper's presentation (figures in appendix, formatting errors), and the method's bad performance in stochastic environments.

**Reviewer Concerns:**

**Concerns addressed by rebuttal**

- Reviewer iyK8 correctly identified that in Table 2, the baseline (e_a, 0)-OGA appeared to outperform the proposed KVDA at 1,000 iterations. In the rebuttal, the authors admitted this was a huge blunder where the column headers were accidentally swapped. With the correction, KVDA outperforms the baseline. This addresses the primary empirical concern of Reviewer iyK8.
- Reviewers VwR2, iyK8 criticized the placement of key results (Figures 2 and 3) in the appendix and general presentation of the paper. The authors have moved these figures to the main text and reorganized Section 3 (Method-first as suggested by Reviewer iyK8).
- The authors clarified scenarios where KVDA might theoretically struggle (incomplete successor sets) in response to Reviewer VwR2's question.

**Still outstanding / partially addressed concerns:**

- Reviewer EN6Y highlighted that the method is limited to deterministic, single-player domains. The extension to stochastic environments performed often worse than baselines, and the method does not apply to multi-player games (due to non-unique V*). While the authors argue their focus is filling the gap for deterministic settings, this remains a limitation on the general applicability of the work compared to standard MCTS improvements.
- Reviewer EN6Y noted the reliance on hand-engineered heuristics to create dense rewards in board games, which aids the difference-based calculation. While the authors argue results are qualitative similar in other MDPs, the reliance on specific reward structures to enable the inference of value differences remains a constraint of the framework.

**Reviewer Scores:**

- Reviewer VwR2: Likely to maintain 6. Their concerns were mostly presentational and have been addressed.
- Reviewer iyK8: Likely to update 4 to 6. Their specific concern was that the baseline appeared superior at high iterations. Since this was a labeling error and the corrected data shows the proposed method winning, the concerns are resolved.
* Reviewer EN6Y: Likely to maintain 4 or move to 6. The reviewer’s fundamental concern regarding the limited scope and stochastic performance is inherent to the method and was acknowledged but not fundamentally fixed by the authors.

---

### Decision · Program_Chairs · 2026-01-26

Accept (Poster)